# Adaptive Coding Emerges in Stabilized Supralinear Networks Trained with Local Plasticity

**Haoyu Albert Wang** [* 1 2 3 †] **Wei P. Dai** [* 1] **Jialun Ma** [1] **Jiawei Zhang** [1 2] **Jinqi Liu** [1] **Mingchen Jiang** [1]
**Mingqing Xiao** [4] **Yansen Wang** [4] **Dongqi Han** [4] **Dongsheng Li** [4] **Yuguo Yu** [1 2 3]

## Abstract

Lateral connections (LCs) are ubiquitous in the cortical circuits. While DL architectures have rich intralayer interactions to support feature selectivity and contextual modulation, explicit excitatory and inhibitory (E-I) LCs remain underexplored and less-justified for encoding models in both DL and visual neuroscience. In this work, we analyze and train stabilized supralinear networks (SSNs) with strong E-I LCs, using local plasticity rules and natural images. We demonstrate that these LCs support a transition between dynamical regimes under different input conditions. During the transition, the network shifts from population coding that extracts features from low-contrast or noisy inputs by recruiting more neurons, to sparse coding at high contrast, utilizing considerably fewer neurons. This reduction in the number of active neurons has been generally associated with lower metabolic demand in previous experiments and models. We find the model showing better robustness and adaptiveness against sparse coding, ICA and other unsupervised models under degraded inputs, but not when LCs are ablated. These results support the role of E-I recurrence in dynamic coding strategies and the design of more adaptive and robust systems with a concrete example in vision.

---

[*] Equal contribution [†] Work done during internship at Microsoft Research Asia, Shanghai. [1] Research Institute of Intelligent Complex Systems, Fudan University, Shanghai, China [2] Institute of Science and Technology for Brain-Inspired Intelligence, Fudan University, Shanghai, China [3] State Key Laboratory of Brain Function and Disorders and MOE Frontiers Center for Brain Science, Institutes of Brain Science, Fudan University, Shanghai, China [4] Microsoft Research Asia, Shanghai, China. Correspondence to: Wei P. Dai <wei-dai@fudan.edu.cn>, Dongqi Han <dongqihan@microsoft.com>, Dongsheng Li <dongsli@microsoft.com>, Yuguo Yu <yuyuguo@fudan.edu.cn>.

*Proceedings of the 43$^{rd}$ International Conference on Machine Learning*, Seoul, South Korea. PMLR 306, 2026. Copyright 2026 by the author(s).

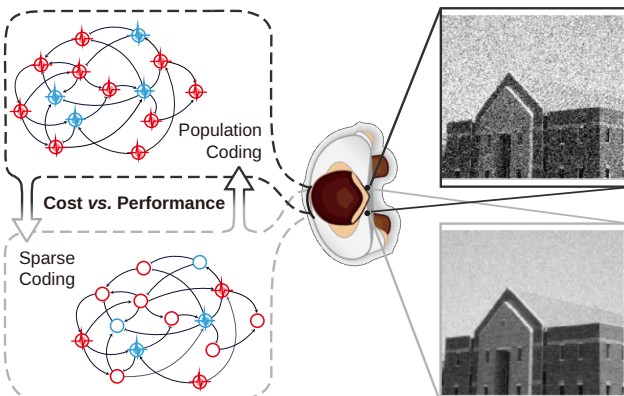

*Figure 1.* **Overview.** Dynamical switch between population coding and sparse coding based on trade-off between performance and cost under different input conditions.

## 1. Introduction

Sifting through the vast literatures of modern deep learning (DL) development and advances in understanding the brain, two gaps between the fields stand out prominently. (i) The brain still outperforms in adaptiveness and robustness under various input conditions. (ii) The brain is organized in fundamentally different manner, not just through the effective deepness (Suzuki et al., 2023), but also through activation function (Geadah et al., 2024) and intralayer recurrency (Park et al., 2021). The first gap in performance could potentially be addressed by closing the second gap in structure along two general pathways in the brain. One is the top-down goal-oriented (*active*) pathway, and another is the bottom-up input-dependent (*passive*) pathway, with the latter pathway largely supported by the explicit intralayer excitatory and inhibitory (E-I) recurrency in the cortical circuits (Lien & Scanziani, 2013).

Modern DL models have also increasingly relied on *implicit* forms of intralayer interactions, e.g., convolution mixing, normalization and attention to patch up these gaps with substantial improvement in task performance. However, they still lag behind the brain in terms of adaptiveness and robustness (Geirhos et al., 2021; Wichmann & Geirhos, 2023; Shen et al., 2025). Meanwhile, increasing literatures suggest

explicit lateral connections (LCs) in the brain play a central role in sensory processing (Adesnik & Scanziani, 2010; Kar et al., 2019; Seijdel et al., 2021) but they are rarely incorporated explicitly into task-performing models, and when they are, their functional role is often left unjustified or treated as a biological embellishment rather than a necessity (Cohen et al., 2022; Zhong et al., 2025) (but see memory studies (Vidal-Saez et al., 2024)). This creates a clear discrepancy between the ubiquitous existence of explicit E-I LCs and their necessary roles in the brain vs. the reluctance of its adoption in artificial systems. Resolving this discrepancy require moving beyond the biological plausibility of including these connections and toward a principled functional interpretation and implementation in task-performing models for different input conditions.

**Problem Setting:** Can the inclusion of explicit E–I lateral connections in neural network significantly improve visual encoding capabilities under different input conditions?

To address this problem, we resort to the stabilized supralinear network (SSN) which has been known to sit well with key response properties in experiment observations though hard to train with strong recurrent excitation (Ahmadian et al., 2013; Echeveste et al., 2020; Soo & Lengyel, 2022). As a preliminary, we theoretically analyzed a two-population SSN and show that the network exhibits supralinear integration (as expected) of weak inputs over an extensive range of parameters, demonstrating a strong capacity to amplify weak feature signals. We further show that when multiple populations in the SSN interact cooperatively, strong inputs can push the network into a regime in which a small subset of highly active neurons strongly inhibits the remaining units, suggesting a mechanism consistent with sparse coding. Next, we successfully trained a large-scale SSN on natural images using only local learning rules. The trained network exhibits a range of properties reminiscent of primary visual cortex (V1), including the emergence of Gabor-shaped receptive fields. Remarkably, the network spontaneously adapts its coding strategy to different input conditions—such as varying contrast levels—transitioning between population coding and sparse coding regimes without any external supervision or explicit inductive bias. Moreover, under non-ideal input conditions, the SSN consistently outperforms classical models such as independent component analysis (ICA(Bell & Sejnowski, 1997)), sparse coding (Olshausen & Field, 1996), SimCLR, two predictive coding models (Rao & Ballard, 1999; Lotter et al., 2017), and a Poisson autoencoder (Vafaii et al., 2024).

Finally, We note that this auto-switching of coding strategies is metabolically efficient since it takes more energy for a silent neuron to become active than driving an active neuron to fire more (Hasenstaub et al., 2010; Yi et al., 2016). For degraded input features during low-contrast or noisy

conditions, more neurons are recruited, each casts its own vote for more robust feature detection with higher metabolic cost. While under proper high-contrast conditions, only a few neurons per visual field are utilized for signalling their preferred feature with higher firing rate rather than using more neurons, reducing metabolic cost Fig.1. Importantly, this transition emerges naturally from the trained network dynamics without any explicit constraint from global loss functions or regularizations on sparsity.

## Contributions

- A biologically plausible unsupervised learning framework for the established SSNs with strong E-I recurrence.

- The emergence of an input-dependent switching of coding strategy from local learning in SSN.

- Empirical evidence that this adaptive strategy improves robustness under degraded inputs vs. alternative unsupervised methods.

## 2. Related Works

**SSNs** model cortical circuits with supralinear transfer functions, strong recurrent excitation, and feedback inhibition in an inhibition-stabilized regime (Ahmadian et al., 2013; Tsodyks et al., 1997; Sanzeni et al., 2020). Foundational SSN work and subsequent extensions have accounted for a broad range of cortical response properties, including contrast-dependent normalization, surround suppression, cross-orientation suppression (Rubin et al., 2015), variability quenching (Hennequin et al., 2018), cell-type-specific effects (Millman et al., 2020), attentional modulation (Lindsay et al., 2020), bistable, oscillatory, persistent, and gamma-band regimes (Kraynyukova & Tchumatchenko, 2018; Holt et al., 2024), as well as spatial and contextual effects (Obeid & Miller, 2025; Wu et al., 2026). Despite this explanatory breadth, contrast-dependent sharpening of orientation tuning has not been studied in SSNs to our knowledge; and SSNs remain difficult to train: strong recurrent excitation combined with expansive, non-saturating single-neuron non-linearities makes them susceptible to dynamical instabilities and runaway excitation. As a result, prior work has often relied on hand-crafted SSN configurations (Rubin et al., 2015; Kraynyukova & Tchumatchenko, 2018), specialized neural growth procedures (Soo & Lengyel, 2022), or heavily constrained, under-parameterized networks (Echeveste et al., 2020), limiting their expressivity as learned models of cortical computation. Here, we address this gap by training SSNs with local plasticity on natural images and analyzing the adaptive coding regimes that emerge from the learned recurrent dynamics, with contrast-dependent sharpening of orientation tuning as one of the examples.

**Sparse coding and ICA** are grounded in efficient coding theory and aim to reduce redundancy through different principles. Sparse coding seeks to minimize entropy by enforcing a small number of active coefficients in an overcomplete basis (Olshausen & Field, 1996), whereas ICA maximizes the joint entropy of a nonlinearly transformed output feature vector(Bell & Sejnowski, 1997). Both frameworks successfully account for the emergence of Gabor-like receptive fields (RFs), as observed experimentally, and have therefore become dominant theoretical paradigms in visual neuroscience for explaining the coding of natural images (Vinje & Gallant, 2000; Hyvärinen & Hoyer, 2001; Klavinskis-Whiting et al., 2025; Hyvärinen & Oja, 2000; King et al., 2013). Despite these successes, experimental evidence suggests that cortical activity is often less sparse than predicted by these models and depends strongly on input conditions (Vinje & Gallant, 2000; Tolhurst et al., 2009). Moreover, large-scale neural recordings indicate that V1 operates in a moderately dense, high-dimensional regime in which many neurons are simultaneously active (Stringer et al., 2019).

**Recurrency and Positioning** Previous studies (Kubilius et al., 2019; Geirhos et al., 2021; Nayebi et al., 2022)) show that adding recurrency (implicitly or abstractly) to models enhance visual task performance for degraded inputs. However their performance still lags behind human (Wichmann & Geirhos, 2023; Shen et al., 2025) and the computational roles of these recurrencies are not well understood (Cohen et al., 2022). Here we build models based on the more biologically-aligned SSN to argue for the necessity of having explicit E-I recurrency from a new perspective – its contribution to an adaptive and metabolically-efficient coding strategy for different input conditions.

# 3. Preliminary: Nonlinear Expansion and Activity Redistribution in SSNs

Over the past two decades, ICA and sparse coding have become the dominant theoretical frameworks for explaining the emergence of receptive fields in V1 and how they encode sensory inputs. In ICA, neuronal responses are modeled using linear dynamics, with nonlinearity arising exclusively through the learning of synaptic weights. As a consequence, ICA admits a trivial state-space structure: for a given stimulus, each neuron's response is uniquely determined, and the system effectively possesses a single fixed point in state space (Fig. 2A,left). Sparse coding extends this framework by introducing mutual inhibitory interactions between neurons. The strength of inhibition is determined by the correlation between receptive fields, giving rise to competitive dynamics in which neurons "explain away" one another. In this setting, the feedforward input defines the admissible region of the state space, while the recurrent inhibitory dynamics iteratively drive population activity toward the origin

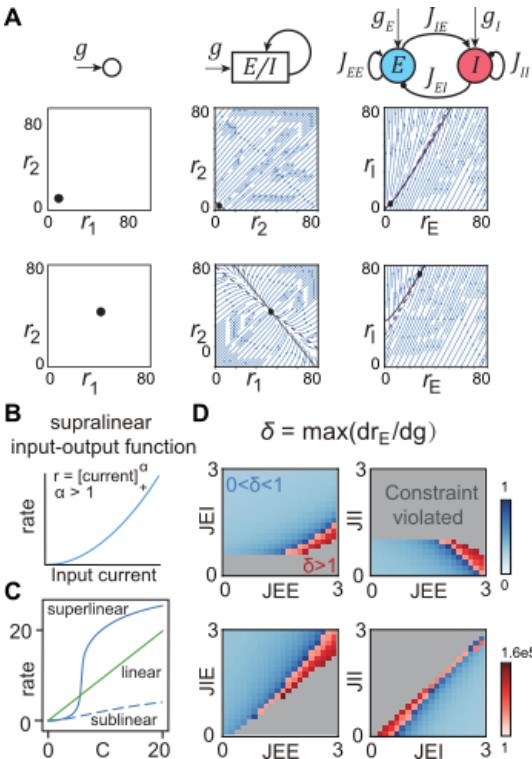

*Figure 2.* **Comparison of ICA, sparse coding and SSN.**
**A.** Schematic comparison of ICA, sparse coding, and SSN models and their phase-space. Phase portraits under weak and strong inputs are shown in the second and third rows. ICA lacks intrinsic dynamics. Sparse coding exhibits input-dependent phase-space boundaries. SSN displays rich dynamical phase structures due to recurrent excitation and feedback inhibition. **B.** Supralinear power-law input–output function of the SSN ($k = 0.04$, $n = 2.0$). **C.** Three types of gain curve: supralinear, linear, and sublinear. **D.** Maximal response gain $\delta = \max(\mathrm{d}r_E/\mathrm{d}g_E)$ as a function of synaptic coupling strengths. Gray regions violate stability constraints; blue and red regions correspond to sublinearity and supralinearity, respectively. Experimental details in the Appendix B.2.1.

(Fig. 2A,middle). When feedforward inputs are normalized to a fixed energy scale, both ICA and sparse coding provide accurate descriptions of V1 responses. Consequently, many subsequent models inspired by these theories adopt similar architectural assumptions, most notably, the lack of excitatory dynamics from explicit excitatry (E) LCs(Zylberberg et al., 2011; King et al., 2013; Brito & Gerstner, 2016).

However, in the biological cortex, E LCs contribute a substantial fraction of the total synaptic current (Lien & Scanziani, 2013). Neglecting these excitatory components leads to an incomplete characterization of V1 dynamics, particularly in regimes where recurrent amplification and feedback inhibition play a critical role. Motivated by this discrepancy, we consider the Stabilized Supralinear Network

(SSN), a recurrent model with explicit E-I LCs, which give rise to a markedly richer and more structured state-space geometry (Fig. 2A,right).

To provide theoretical intuition for our use of the SSN, this section highlights two key properties that fundamentally distinguish it from classical ICA and sparse coding models. First, the SSN exhibits a nonlinear expansion regime in which neural responses grow faster than linearly with input strength (Sec. 3.1). Second, the SSN supports a dynamic transition between sparse and population coding regimes, enabling flexible redistribution of activity across neurons as input statistics change (Sec. 3.2). Together, these features underscore the necessity of E-I LCs for capturing the full dynamical repertoire of cortical computation.

We consider a general SSN composed of interacting E and inhibitory (I) neuronal populations. The population-averaged firing-rate dynamics are described by

$$\tau_E \frac{\mathrm{d}\mathbf{r}_E}{\mathrm{d}t} = -\mathbf{r}_E + k \left[ \mathbf{J}_{EE}\mathbf{r}_E - \mathbf{J}_{EI}\mathbf{r}_I + \mathbf{g}_E \right]_+^{\alpha_E}, \quad (1)$$

$$\tau_I \frac{\mathrm{d}\mathbf{r}_I}{\mathrm{d}t} = -\mathbf{r}_I + k \left[ \mathbf{J}_{IE}\mathbf{r}_E - \mathbf{J}_{II}\mathbf{r}_I + \mathbf{g}_I \right]_+^{\alpha_I}, \quad (2)$$

where $\mathbf{r}_E \in \mathbb{R}^{N_E}$ and $\mathbf{r}_I \in \mathbb{R}^{N_I}$ denote the firing-rate vectors of the excitatory and inhibitory populations, respectively. The constants $\tau_E$ and $\tau_I$ are the corresponding neuronal time constants. The matrices $\mathbf{J}_{XY}$ represent synaptic coupling from population $Y$ to population $X$, with $X, Y \in \{E, I\}$. The vectors $\mathbf{g}_E$ and $\mathbf{g}_I$ denote external input (sensory stimuli) drives to the excitatory and inhibitory populations, respectively.

The nonlinear input–output transformation is given by a rectified power-law function, where $[\cdot]_+$ denotes rectification (Fig 2B), $\alpha_E$ and $\alpha_I$ are the supralinear exponents (set to 2.0 unless stated otherwise (Rubin et al., 2015)), and $k$ is a global gain parameter controlling the overall amplification of input currents. This formulation captures the defining features of the SSN, namely strong recurrent excitation stabilized by feedback inhibition, and serves as a general framework for modeling cortical dynamics.

### 3.1. Nonlinear Expansion Regime

As discussed above, both ICA and sparse coding exhibit an approximately linear dependence of neuronal responses on input strength, which necessitates explicit normalization of the input. In contrast, biological systems operate in environments that are inherently noisy and characterized by dynamic input intensities (or contrasts after retinal transformation) across magnitudes. To function robustly under such conditions, cortical circuits must adapt to changes in input strength without relying on strict normalization. Consequently, neurons in V1 are expected to exhibit a response regime that goes beyond linear scaling, namely a

*nonlinear expansion* in which the gain exceeds unity, i.e., $\mathrm{d}\mathbf{r}_E/\mathrm{d}\mathbf{g}_E > 1$ over a finite range of input strength.

To understand how such nonlinear expansion behavior emerges in recurrent cortical circuits, we analyze the SSN in its minimal form, consisting of a single E and I population, i.e., Eqs. 1 and 2 reduced to their scalar form. To ensure that the network operates in a regime of strong feedback inhibition—where stable fixed points are maintained across a broad range of external input strengths $g_E$ and $g_I$—we impose two constraints on the connectivity parameters. First, we require

$$\det(\mathbf{J}) = -J_{EE}J_{II} + J_{IE}J_{EI} > 0, \quad (3)$$

which ensures the existence and stability of fixed points in the coupled E–I system. Second, we impose

$$J_{II}g_E - J_{EI}g_I < 0, \quad (4)$$

which guarantees that the population activity exhibits saturation rather than unbounded growth with the external drive(Ahmadian et al., 2013). Together, these conditions characterize an inhibition-stabilized regime in which feedback inhibition dynamically counterbalances recurrent excitation.

The system can be analytically analyzed using a characteristic function that reduces the dynamics to an effective one-dimensional description (Kraynyukova & Tchumatchenko, 2018). We modify this function to explicitly account for the effect of the gain parameter $k$, as follows:

$$F(z) = J_{EE} k [z]_+^{\alpha_E} - z + g_E - J_{EI}k [\omega]_+^{\alpha_I},$$
$$\omega = J_{EI}^{-1} \left( \det(\mathbf{J}) k [z]_+^{\alpha_E} + J_{II}(z - g_E) + J_{EI}g_I \right). \quad (5)$$

where $z, \omega$ denote the total input current to the E and I population, respectively. The zero crossings of $F(z)$ correspond exactly to the steady-state solutions of the full system defined by Eqs. (1)and (2). Specifically, $z$ is a root of $F(z)$, with the firing rates given by

$$r_E = k [z]_+^{\alpha_E}, \quad r_I = k [\omega]_+^{\alpha_I}. \quad (6)$$

Thus, the nonlinear expansion behavior of $r_E$ can be fully characterized by the solutions of Eqs. (5) and (6). Requiring $\mathrm{d}r_E/\mathrm{d}g_E > 1$ then yields the following inequality (Appendix A):

$$2kz(1+J_{EE}) - 1 > \frac{2k}{J_{EI}}$$
$$\times \left( \det(\mathbf{J}) kz^2 + J_{II}z + (J_{EI} - J_{II})g \right) \quad (7)$$
$$\times \left( 2kz \left( \det(\mathbf{J}) + J_{EI} - J_{II} \right) + J_{II} \right).$$

Although the above inequality does not admit a simple quantitative characterization of the relationships among the synaptic weights, several qualitative insights can nevertheless be obtained. For example, from the left-hand side of the

inequality, increasing $J_{EE}$ tends to promote its satisfaction; in contrast, the right-hand side can be approximated by a first-order term in $J_{EI}$, indicating that smaller values of $J_{EI}$ are favorable.

Using numerical analysis, we mapped the regions of supra-linear expansion (red) and sublinear expansion (blue) in the parameter space spanned by $J_{EE}$, $J_{EI}$, $J_{IE}$, and $J_{II}$ (Fig. 2D). The resulting phase diagram is consistent with our theoretical predictions, and the distinct response regimes can be directly observed in (Fig. 2C). Notably, the nonlinear expansion regime occupies only a small fraction of the overall phase space and lies close to the boundary of dynamical instability, which explains why SSNs are notoriously difficult to train.

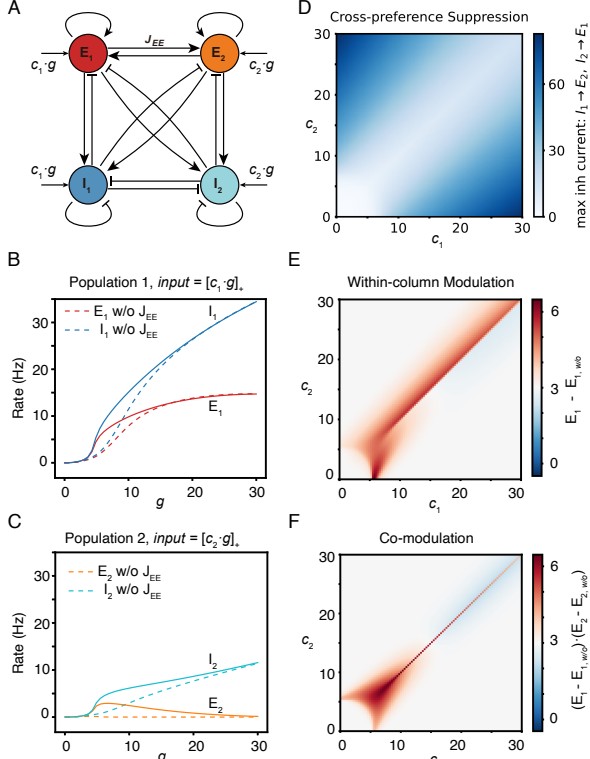

*Figure 3.* **Activity Redistribution based on Feature Signal.**
**A.** Connectivity diagram for two pairs of E-I SSN. **B and C.** An example of gain curves with (solid) and without (dashed) recurrent excitation between populations with feature preference, $c_2 = 0.7 * c_1$.
**D.** The difference in E responses of the two populations scaled by $c_1 - c_2$. **E.** The modulation effect through difference in $E_1$'s response of a single population with vs. without recurrent excitation ($E_2$ is diagonally symmetric to $E_1$). **F.** Dot-product of two populations' modulation effect (as shown in E.). Experimental details in the Appendix B.2.2.

### 3.2. Activity Redistribution based on Feature Signal

To further understand how nonlinear expansion works with feature extraction among multiple populations that have

different preferences like in cortical columns, we extend out analysis to two interacting E and I population pairs (Fig.3A). We abstract the feed-forward receptive fields of each E-I pair by a feature signal strength $c_i(g), i = 1, 2$ that depends on the input $g$, allowing the two pairs to respond differentially to the same stimulus.

As an example, we show the gain curves of the two population (Fig. 3B and C) when Eq. 7 is met and the preferential difference is moderate, $c_2 = 0.7c_1$. After both populations jointly enter the nonlinear expansion regime, a clear redistribution of activity emerges: recurrent excitation amplifies both populations at weak input, but selectively suppresses the population with weaker feature support as input strength increases (solid vs. dashed curves). Importantly, strong feedback inhibition preserves the winner-takes-all (WTA) regime at high input, preventing runaway excitation. However, examining closely across pairs of preferential differences $(c_1, c_2)$ with fixed $g$ reveals weaker effects of cross-preference suppression, but only for similar preferences near the diagonal (Fig. 3D. This is quantified as the ratio of difference in population activity (normalized by $\max(E_1, E_2)$) between the network with recurrent excitation and without.

Within a population of fixed feature preference, weak feature signals are strongly amplified once they cross the sensitivity threshold (around $c_1 = 5$; Fig. 3E). This amplification is maximal when $c_1 \gg c_2$, reflecting high confidence in feature 1 despite degraded input, and supports robust feature detection. As evidence for feature 2 increases, this effect diminishes, though the combined feature signal remains amplified at low input strength.

For larger $c_1$, amplification above the diagonal arises from recurrent excitation driven by increasing $c_2$, whereas saturation (hints of blue) below the diagonal reflects stronger feedback inhibition, also visible in Fig. 3B.

To summarize, recurrent excitation co-modulates populations with different feature preferences by jointly amplifying weak signals while preserving suppression of the less-supported population when feature signals are strong or highly imbalanced elsewhere (Fig. 3F).

## 4. Self-Emergent Dynamic Coding Strategies in a Learned SSN

Although the strong nonlinearity of the SSN makes direct analytical treatment of high-dimensional systems intractable, this section demonstrates—through computational simulations—that the two dynamical properties proposed in Sec. 3, namely nonlinear expansion and dynamic coding strategies, are reachable through biologically plausible synaptic plasticity. These results establish that the theoretical regimes identified in simplified models can in fact emerge through learning in large-scale recurrent networks.

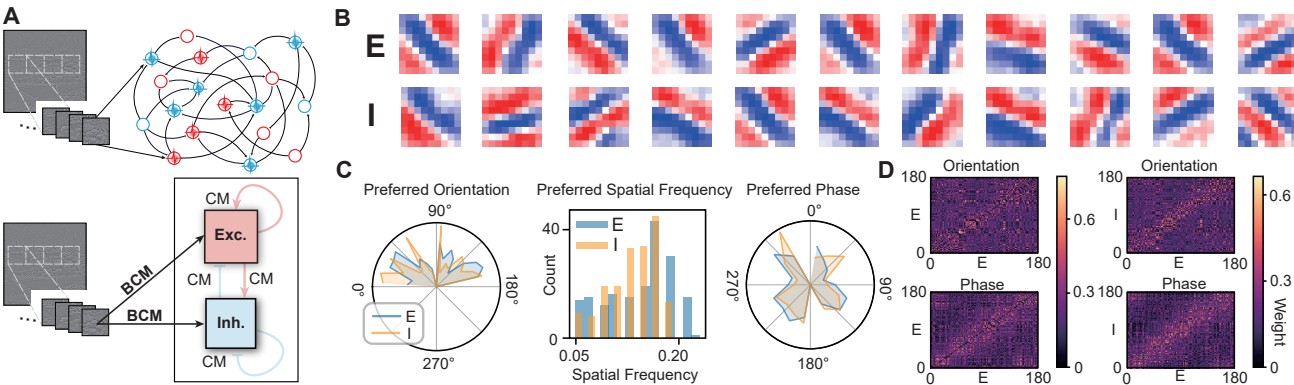

*Figure 4.* **SSN learning with local rules. A.** Training pipeline: whitened natural-image patches drive E and I populations; feedforward weights follow BCM and recurrent weights follow CM. **B.** Example learned receptive fields (STA). **C.** Orientation, spatial-frequency, and phase preference distributions in E and I. **D.** E–I weight matrix with neurons sorted by preferred orientation (top) or phase (bottom); rows: I, columns: E.

Accordingly, in Sec. 4.1, we introduce the overall model architecture and the synaptic plasticity rules used for training, then we present the learned receptive fields, showing that the trained SSN develops V1-like receptive field structures. Remarkably, as reported in Sec. 4.2, we find that under grating stimuli, many neurons in the learned SSN exhibit sharpened tuning curves, a qualitative difference from SSNs with hand-designed connectivity. Furthermore, when probed with natural images at varying contrast levels, the trained SSN spontaneously develops dynamic coding strategies, flexibly transitioning between sparse and population-based representations in response to changes in input statistics. Finally, in Sec. 4.3, we trained a large-scale SSN on CIFAR-10 and compared it with sparse coding, ICA, and other unsupervised models, demonstrating its superior representational performance under degraded input conditions.

### 4.1. Model Setup and Learning Rules

Following standard practice in sparse coding models, we whitened 20 original natural images to equalize the spectral power across spatial frequencies (Fig. 4A), mimicking preprocessing in the lateral geniculate nucleus(Olshausen & Field, 1996). In each trial, all E and I neurons receive visual input from 1024 randomly sampled $9 \times 9$ image patches drawn from this set of 20 whitened natural images.

The SSN consists of 180 excitatory and 180 inhibitory neurons(Fig. 4A), matching the canonical configuration adopted in prior studies(Rubin et al., 2015). These neurons form a recurrently connected network whose firing-rate dynamics are described in Eq. 1 and Eq. 2. Both E and I populations are directly driven by natural image inputs, i.e., $g_{E/I} = W_{E/I}^{\mathrm{FF}}$.

Synaptic plasticity is governed by two classes of Hebbian-like learning rules: the Bienenstock-Cooper-Munro (BCM) rule(Bienenstock et al., 1982) and a correlation-measuring (CM) rule(King et al., 2013).All feedforward connections are trained using the BCM rule, a well-established mech-

anism in visual neuroscience for receptive field formation and orientation selectivity. In contrast, all lateral recurrent connections (E–E, E–I, I–E, and I–I) are updated using a correlation-measuring (CM) rule, allowing recurrent circuitry to capture correlations between neuronal responses and input-driven activity.

This separation of learning rules is essential for stabilizing recurrent excitation while enhancing representational efficiency. BCM drives self-organized feature learning in the feedforward pathway, whereas CM shapes structured recurrent interactions. In particular, CM learning among excitatory neurons supports mutual amplification under weak inputs, while at higher input strengths—when the network enters the inhibition-stabilized regime —correlated excitatory activity is transferred to inhibitory neurons and suppressed via E–I–E feedback loops. This mechanism reduces redundancy and enables sparse, efficient population coding with minimal hyperparameter tuning.

Synaptic updates are applied after every 1 s of stimulus presentation, with learning signals averaged over a batch of 1024 input samples. Denoting by $x_j$ and $y_i$ the mean firing rates of the presynaptic neuron $j$ and postsynaptic neuron $i$, respectively, averaged over a single image presentation, the synaptic weight changes $\Delta W_{ij}^{K \leftarrow K^*}$ are computed as

**Feedforward (BCM):**$\Delta W_{ij} \propto y_i \, x_j \, (y_i - \theta_i),$

**Recurrent (EE, EI, IE, II; CM):** $\qquad$ (8)
$$\Delta W_{ij} \propto y_i \, x_j - \langle y_i \rangle \langle x_j \rangle \, (1 + W_{ij}).$$

where $K, K^* \in \{E, I\}$ denote the post- and presynaptic neuron types, respectively, $\theta_i$ is the BCM sliding threshold, and $\langle \cdot \rangle$ denotes a batch average over image patches.

All learning rates are set to $10^{-4}$. To mitigate training instabilities commonly encountered in SSNs, we additionally impose weight normalization to ensure that the total incoming synaptic strength to each neuron remains constant. Specifically, after each update, synaptic weights are renormalized

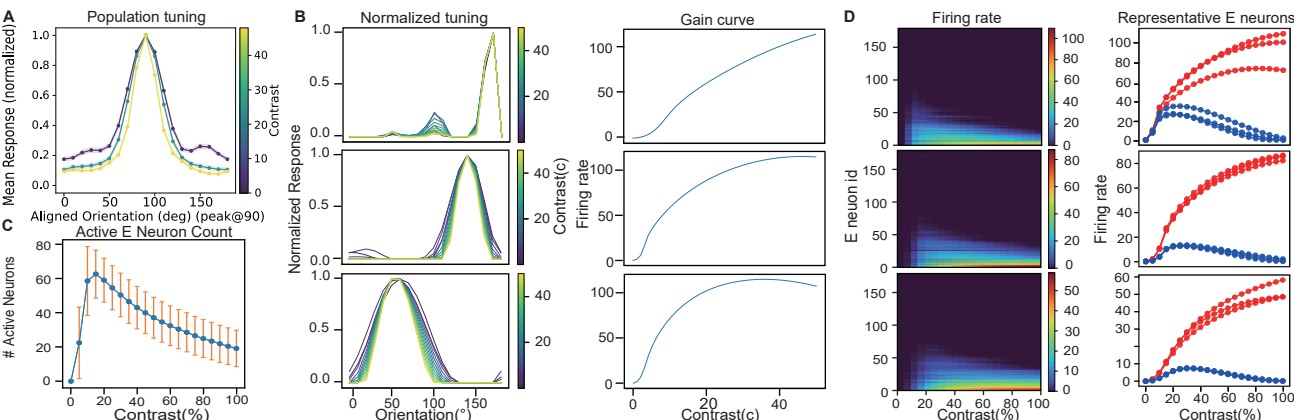

*Figure 5.* **Auto-switching Coding Strategy A.** Population averaged tuning curves under different contrast aligned at the optimal orientation. **B.** Example neurons for contrast-dependent contrast-invariant and sharpening tuning curves and contrast gain curves. **C.** Active number of neurons for each natural image stimulus (mean + std. errorbar) over differnet contrast levels. **D.** Firing rate heatmap of neurons in response for example patches of natural image under different contrast (sorted by the highest) with selected contrast gain curves in the population.

according to

$$\mathbf{W} \leftarrow \mathbf{W} \frac{W_{\text{init}}}{\sum_j W_{ij}}, \qquad (9)$$

where $W_{\text{init}}$ is the initial total input weight for the corresponding connection type. The target total synaptic strengths are set to 3.75 for E–E, 1.86 for E–I connections, 3.13 for I–E connections, and 1.09 for I–I connections. This normalization scheme stabilizes learning while preserving relative synaptic structure.

After exposure to thousands of image patches drawn from whitened natural images, both E and I neurons developed Gabor-like receptive fields with diverse orientations, spatial frequencies, and phases (Fig. 4B), consistent with classical findings from ICA (Bell & Sejnowski, 1997) and sparse coding models(Olshausen & Field, 1996). Fig. 4C summarizes the statistical distributions of preferred orientation, spatial frequency, and phase across the E and I populations. These distributions are approximately uniform and highly similar between E and I neurons, indicating a strong degree of E/I cotuning, a hallmark of response property observed in V1 (Okun & Lampl, 2008).

We further analyzed the learned recurrent connectivity structure, focusing on the E–E and E–I sub-networks. In Fig. 4D, we reorder E and I neurons by their preferred orientation and phase, and visualize the corresponding connectivity matrices. After this ordering, the E–I connectivity exhibits a pronounced diagonal band (Fig. 4D, right column), indicating strong synaptic coupling between E and I neurons with similar stimulus preferences. This stimulus-aligned E–I coupling provides the circuit substrate for the emergence of sparse coding under strong drive, as feature-matched inhibition can efficiently suppress redundant excitatory activity through E–I feedback loops. In contrast, the diagonal structure in the E–E connectivity is noticeably weaker(Fig. 4D, left column). This suggests that while E neurons still pref-

erentially connect to others with similar tuning, the recurrent excitation is more distributed and thus tends to recruit a broader subset of neurons. Such broader recurrent recruitment is consistent with a population-coding regime, in which responses are supported by distributed excitation rather than sharply selective inhibition.

### 4.2. Self-Emergent Dynamic Coding Strategies

To characterize the response properties of the trained SSN, we evaluated the network under two stimulus regimes: artificial grating stimuli and natural scenes. We first used a comprehensive grating set consisting of 18 orientations spanning $[0°, 180°)$, 16 spatial frequencies ranging from 0.05 to 0.35, and 16 phases spanning $[0°, 360°)$. For each neuron, we identified the optimal stimulus defined by the parameter combination that elicited the maximal response. Holding these preferred parameters fixed, we then probed neuronal responses using gratings of varying contrast to examine contrast-dependent response modulation.

Figure 5A shows the population-averaged tuning curves after aligning neurons by their preferred orientation. The resulting tuning profiles and their half-width at half-maximum are quantitatively consistent with experimental measurements in V1(Ringach et al., 2002), supporting the biological plausibility of the learned network. Notably, when responses were measured across different contrast levels, we observed that many neurons exhibited pronounced contrast-dependent sharpening of their tuning curves.

**Contrast-dependent sharpening** The neural mechanisms underlying contrast-dependent sharpening of orientation tuning have long been debated (Finn et al., 2007; Dai et al., 2018). This issue is often framed in terms of contrast invariance: as feedforward input from the lateral geniculate nucleus (LGN) to V1 increases with contrast, classical feed-

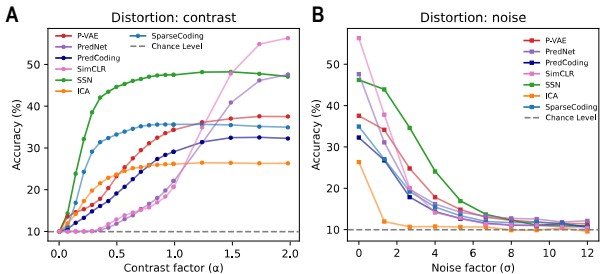

*Figure 6.* **Comparison with Degraded Inputs.** CIFAR10 classification under varying contrast and noise levels. Classification accuracy is shown for SSN, ICA, sparse coding, P-VAE, PredNet, predictive coding and SimCLR. The model size is limited to approximately 14,400 latent neurons for each.**(A)** Under low-contrast conditions, SSN consistently outperforms the other models, suggesting that its population-coding strategy improves robustness to contrast distortion. **(B)** Under increasing noise levels, SSN shows a slower decline in accuracy than the other models, indicating greater stability against noise distortion.

forward models predict a widening of tuning curves—an "iceberg" effect—that contradicts experimental observations (Sclar & Freeman, 1982). Here, we demonstrate that neurons operating in the learned SSN regime naturally exhibit contrast-dependent sharpening (i.e., increased orientation selectivity with contrast; Fig. 5B, the 2nd and 3rd rows) or contrast-invariant tuning (i.e., selectivity does not decrease with contrast; Fig. 5B, the 1st row), in agreement with physiological findings in V1 (Johnson et al., 2008; Li et al., 2012). Importantly, we attribute this phenomenon not to fine-tuned parameters, but to an automatic switching between metabolically cost-effective coding strategies. At low contrast, the network operates in a population-coding regime, recruiting a larger number of neurons to represent weak orientation signals. As contrast increases, the network transitions to a sparse-coding regime in which fewer, more selectively tuned neurons dominate the response.

**Dynamic coding strategies in natural images** To move beyond artificial stimuli, we probed the trained SSN with natural scenes. We sampled 500 natural image patches and varied contrast from 0% to 100%, measuring population representations at each level. The network spontaneously expresses distinct coding regimes across contrast (Fig. 5D).

Fig. 5D (left) shows three example patches, with neurons sorted by their 100%-contrast responses. As contrast rises, the active set expands rapidly (peaking near $\sim 20\%$) and then contracts, leaving only the strongest responders at high contrast. The right panel highlights two representative neuron classes (cf. E1/E2 in Fig. 3): neurons tuned to the patch's dominant features (red) strengthen with contrast, while neurons driven by non-preferred features (blue) are progressively suppressed.

These dynamics reflect a transition from population cod-

ing to sparse coding with increasing contrast. We quantify this in Fig. 5C by counting neurons with firing rates above 1 Hz at each contrast. The count rises sharply at low-to-intermediate contrast (population coding) and then falls at high contrast, indicating sparse representations dominated by a small, highly selective subset.

Given that recruiting silent neurons is metabolically more costly relative to increasing rates of already-active neurons (Hasenstaub et al., 2010; Yi et al., 2016), this transition supports metabolically cost-effective computation: under degraded inputs, the network recruits more neurons to boost robustness (Sec 4.3) despite higher cost; under ideal inputs, it relies on a few strongly active neurons to encode unequivocal features efficiently.

### 4.3. Task Performance Comparison with Other Models

To evaluate the functional capabilities of the SSN, we compared the robustness of its learned representations with those of several baseline models, including ICA, sparse coding, Poisson VAE (P-VAE) (Vafaii et al.), Predictive Neural Network (PredNet) (Rane et al., 2020), Predictive Coding (PredCoding) (Rao & Ballard, 1999), and SimCLR (Chen et al., 2020), under varying contrast and noise conditions. Specifically, we trained a large-scale SSN with 14,400 neurons on whitened CIFAR10 images, using a local connection topology and a training procedure similar to that described in Sec. 4.1 (details in Appendix B.2.4). We then trained only an auxiliary classifier to perform CIFAR10 classification using the representations produced by each model under different perturbation conditions. The resulting classification performance is summarized in Fig. 6. We found that under low-contrast and noise conditions, where population coding is advantageous, the SSN consistently outperformed the other models. In addition, we compared the performance of the network trained in Sec. 4.1 on an orientation-classification task and observed a similar trend (details in B.2.3). Ablation experiments further showed that removing the $E \rightarrow E$ connections impaired population coding and substantially degraded performance under low-contrast conditions, whereas removing the $I \rightarrow E$ connections prevented the network from forming sparse representations and led to unstable activity across contrast levels. Together, these results suggest that lateral recurrent connectivity enables the SSN to recruit larger neural populations under degraded input conditions, thereby supporting more robust and stable representations across diverse sensory environments and conferring a functional advantage when signal quality fluctuates.

## 5. Conclusion

We showed that SSN with explicit E-I LCs can learn to flexibly switch between population coding and sparse coding as

a function of input conditions. Through analysis and unsupervised training with local, biologically plausible learning rules, we identified a dynamical regime transition that enables robust feature amplification under weak or degraded inputs and efficient, low-cost representations under strong inputs – without explicit sparsity constraints. Compared to classical sparse coding and ICA, the learned SSN achieves more metabolically cost-effective encoding while retaining key cortical response properties, including Gabor-like receptive fields, E/I cotuning, and contrast-dependent sharpening that match with experimental observations.

Beyond demonstrating a trainable SSN, we propose that explicit E–I LCs are not merely stabilizing or biological details, but a circuit mechanism that supports adaptive coding strategies by reshaping population dynamics to balance metabolic cost and performance. This perspective helps reconcile why cortical activity is neither uniformly sparse nor dense, but state-dependent and input-adaptive. Finally, the work suggests that E–I structures can endow artificial systems with principled adaptability and robustness, offering a concrete bridge between efficient coding, circuit dynamics, and the design of more flexible neural networks.

**Limitations** The model omits biological dynamics at inference time, including short-term synaptic plasticity and spike-frequency adaptation, affecting energy functions for calculating the exact metabolic cost. Here, we simplify it by using the number of active neurons as a proxy (Yi et al., 2016; Hasenstaub et al., 2010), which does not translate to computational cost on conventional hardware. That said, whether the observed adaptive coding strategy could yield practical benefits in neuromorphic systems (Navaridas et al. 2015) remains to be investigated in future work. Empirically, we did not test robustness to occlusion or colored (temporally correlated) noise, nor evaluate performance on naturalistic movies, where additional temporal structure might expose failure modes.

## Impact Statement

This paper presents work whose goal is to advance the field of neural computation. There are many potential societal consequences of our work, none which we feel must be specifically highlighted here.

## Acknowledgment

We thank the support of Science and Technology Innovation 2030 - Brain Science and Brain-Inspired Intelligence Project (2021ZD0201300), the National Natural Science Foundation of China (92570202), Shanghai Municipal Science and Technology (24JS2810400). The authors thank the Shanghai Institute for Mathematics and Interdisciplinary Sciences (SIMIS) for their financial support (SIMISID-2025-NC). The computations in this research were performed using the CFFF platform of Fudan University.

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

## A. Appendix: Implicit Solution Branch and Sensitivity Analysis

In this appendix, we provide a complete and self-contained derivation of the implicit solution branch $z = z(g)$ of Eq. 5 in the main text, together with closed-form expressions for the first derivatives with respect to the external drive $g$. We further characterize the conditions under which the gain $dz/dg$ exceeds unity .

### A.1. Model reduction and implicit formulation

We start from the scalar fixed-point equation

$$F_k(z) = J_{EE} \, k \, (z)_+^{\alpha_E} - z + g_E$$
$$- J_{EI} \, k \left( J_{EI}^{-1} \Big( \det(\mathbf{J}) \, k \, (z)_+^{\alpha_E} + J_{II}(z - g_E) + J_{EI} g_I \Big) \right)_+^{\alpha_I}, \tag{10}$$

where $\det(\mathbf{J}) = -J_{EE} J_{II} + J_{EI} J_{IE}$. Throughout this appendix, we impose the following assumptions: (i) all arguments of the rectifying nonlinearity are strictly positive (Because we focus exclusively on the smallest positive solution for $z$, motivated by the observation that neural representations typically emerge from a low-activity regime dominated by spontaneous activity, and thus dynamically evolve to the first stable positive solution of $z$); (ii) $\alpha_E = \alpha_I = 2$; (iii) $g_E = g_I = g$. Under these assumptions, rectification can be dropped and the equation reduces to an algebraic implicit relation $F(z, g) = 0$, where

$$F(z, g) = J_{EE} \, k \, z^2 - z + g - J_{EI} \, k \, S(z, g)^2. \tag{11}$$

Here we define

$$S(z, g) = \frac{\det(\mathbf{J}) \, k}{J_{EI}} z^2 + \frac{J_{II}}{J_{EI}} z + \frac{J_{EI} - J_{II}}{J_{EI}} g. \tag{12}$$

We are interested in the *first positive root* $z(g) > 0$ of $F(z, g) = 0$, assumed to be a simple root, i.e., $\partial F / \partial z \neq 0$, so that the implicit function theorem applies.

### A.2. First derivative $dz/dg$

Differentiating $F(z(g), g) = 0$ with respect to $g$ yields

$$\frac{dz}{dg} = -\frac{F_g}{F_z}, \tag{13}$$

where subscripts denote partial derivatives. A direct calculation gives

$$F_g = 1 - 2k \, (J_{EI} - J_{II}) \, S(z, g), \tag{14}$$
$$F_z = 2J_{EE} k z - 1 - 2k \, S(z, g) \, Q(z), \tag{15}$$

with

$$Q(z) := 2 \det(\mathbf{J}) \, k \, z + J_{II}. \tag{16}$$

Substituting Eqs. (14)–(15) into Eq. (13) yields an explicit expression for the gain $dz/dg$ along the selected solution branch.

### A.3. Condition for superlinear gain

We now characterize when the excitatory rate gain exceeds unity, $dr_E/dg > 1$. On the positive branch $z > 0$ with exponent $\alpha_E = 2$, the rate is $r_E(z) = kz^2$, hence by the chain rule,

$$\frac{dr_E}{dg} = \frac{dr_E}{dz} \frac{dz}{dg} = (2kz) \frac{dz}{dg}. \tag{17}$$

Therefore,

$$\frac{dr_E}{dg} > 1 \quad \Longleftrightarrow \quad \frac{dz}{dg} > \frac{1}{2kz}, \tag{18}$$

. Using Eq. (13), $dz/dg = -F_g/F_z$, this inequality is equivalent to

$$\frac{dz}{dg} > \frac{1}{2kz} \quad \Longleftrightarrow \quad -\frac{F_g}{F_z} - \frac{1}{2kz} > 0 \quad \Longleftrightarrow \quad F_z(2kzF_g + F_z) < 0, \tag{19}$$

.

Using Eqs. (14)–(15), we obtain

$$\begin{aligned}
2kzF_g + F_z &= 2kz\Big(1 - 2k(J_{EI} - J_{II})S\Big) + \Big(2J_{EE}kz - 1 - 2kS\,Q\Big) \\
&= 2kz(1 + J_{EE}) - 1 - 4k^2z(J_{EI} - J_{II})S - 2kS\,Q \\
&= 2kz(1 + J_{EE}) - 1 - 2kS\Big(Q + 2kz(J_{EI} - J_{II})\Big),
\end{aligned} \tag{20}$$

. Hence, on the positive-root branch,

$$\boxed{\begin{aligned}
\frac{dr_E}{dg} > 1 \iff \Big(2J_{EE}kz - 1 - 2kS\,Q\Big) \\
\times (2kzF_g + F_z) < 0,
\end{aligned}} \tag{21}$$

with $F_g$ and $F_z$ given in Eqs. (14)–(15).

### A.3.1. SIGN OF $F_z$ ON THE FIRST POSITIVE-ROOT BRANCH

We first characterize a simple range of $g$ for which $F(0, g) > 0$ holds, and then show that on the branch where $z(g) > 0$ is the *first* positive zero of $F(\cdot, g)$, one necessarily has $F_z(z(g), g) < 0$ (away from fold points).

**A sufficient range for $F(0, g) > 0$.** Evaluating (5) at $z = 0$ gives

$$F(0, g) = g - J_{EI}k\,S(0, g)^2. \tag{22}$$

Using

$$S(0, g) = \frac{J_{II}(0 - g) + J_{EI}g}{J_{EI}} = \frac{(J_{EI} - J_{II})g}{J_{EI}}, \tag{23}$$

we obtain

$$F(0, g) = g - \frac{k(J_{EI} - J_{II})^2}{J_{EI}}\,g^2. \tag{24}$$

Hence,

$$F(0, g) > 0 \quad \Longleftrightarrow \quad g\left(1 - \frac{k(J_{EI} - J_{II})^2}{J_{EI}}g\right) > 0, \tag{25}$$

and, for $g > 0$, it suffices to require

$$0 < g < \frac{J_{EI}}{k\,(J_{EI} - J_{II})^2}. \tag{26}$$

**Scale separation.** In our simulations, the difference $(J_{EI} - J_{II})$ is empirically small, and the gain parameter $k$ is also small. As a consequence, the upper bound in Eq. (26) can become very large, implying that the condition $F(0, g) > 0$ is satisfied throughout the entire range of input strengths $g$ explored in our simulations. For instance, with representative parameter values $J_{EI} = 1.3$, $J_{II} = 1.0$, and $k = 0.04$, the resulting upper bound on $g$ is about 360, larger than the maximal $g$ used in the simulations, and therefore does not constitute an active constraint in practice.

**Implication for the sign of $F_z$ at the first positive zero.** Fix any $g$ such that $F(0, g) > 0$. Assume there exists a *first* positive zero $z(g) > 0$ of $F(\cdot, g)$, i.e.,

$$F(z(g), g) = 0, \qquad F(z, g) > 0 \text{ for all } 0 < z < z(g). \tag{27}$$

If this zero is *simple* (i.e., $F_z(z(g), g) \neq 0$), then necessarily

$$F_z(z(g), g) < 0. \tag{28}$$

*Proof.* Let $f(z) := F(z, g)$ with $g$ fixed. Since $f(z(g)) = 0$ and $f(z) > 0$ for all $0 < z < z(g)$, the function reaches the value 0 at $z(g)$ while remaining strictly positive immediately to the left. If $f'(z(g)) > 0$, then for sufficiently small $\varepsilon > 0$ we would have $f(z(g) - \varepsilon) < 0$ (moving left from a point with positive slope decreases the function below zero), contradicting (27). Thus $f'(z(g)) \leq 0$. Because the root is simple, $f'(z(g)) \neq 0$, hence $f'(z(g)) < 0$, i.e., $F_z(z(g), g) < 0$. $\square$

### A.3.2. CONDITION FOR NONLINEAR EXPANSION: $dr_E/dg > 1$

From Eq. (21) and the fact that on the first positive-root branch $F_z < 0$, the condition $dr_E/dg > 1$ is equivalent to

$$2kzF_g + F_z > 0. \tag{29}$$

Substituting the explicit expressions

$$F_g = 1 - 2k(J_{EI} - J_{II})S, \qquad F_z = 2J_{EE}kz - 1 - 2kS\,Q, \quad Q := 2\det(\mathbf{J})kz + J_{II},$$

we obtain

$$2kzF_g + F_z = 2kz(1 + J_{EE}) - 1 - 2kS\Big(Q + 2kz(J_{EI} - J_{II})\Big). \tag{30}$$

Using the definition

$$S(z, g) = \frac{\det(\mathbf{J})\,kz^2 + J_{II}z + (J_{EI} - J_{II})g}{J_{EI}}, \tag{31}$$

and substituting $Q = 2\det(\mathbf{J})kz + J_{II}$, the condition $2kzF_g + F_z > 0$ can be written as

$$2kz(1 + J_{EE}) - 1 > \frac{2k}{J_{EI}}\Big(\det(\mathbf{J})\,kz^2 + J_{II}z + (J_{EI} - J_{II})g\Big)\Big(2kz\big(\det(\mathbf{J}) + J_{EI} - J_{II}\big) + J_{II}\Big). \tag{32}$$

Equivalently, collecting all positive contributions on the left-hand side and all suppressive terms on the right-hand side, the nonlinear expansion condition $dr_E/dg > 1$ reads

$$\boxed{2kz(1 + J_{EE}) - 1 \;>\; \frac{2k}{J_{EI}}\Big(\det(\mathbf{J})\,kz^2 + J_{II}z + (J_{EI} - J_{II})g\Big)\Big(2kz(\det(\mathbf{J}) + J_{EI} - J_{II}) + J_{II}\Big).} \tag{33}$$

# B. Appendix: Methods

### B.1. Measurements

**Orientation Tuning Curves.** We measure the responses of each neuron to gratings of 18 orientations, 16 phases and over 16 octaves of spatial frequencies. Then, we fix the phase and spatial frequency for the optimal orienation response, which is defined as the preferred orientation, and plot its tuning curve.

**Contrast Gain Curve.** To obtain the contrast gain for each neuron, we vary the contrast over 5%, 10%, 15%, 20%, 25%, 30%, 35%, 40%, 45% , 50%, 55%, 60%, 65%, 70%,75.0%, 80%, 85%, 90%, 95% and 100% and record each neuron's response.

### B.2. Experiment Setup

### B.2.1. FIG. 2D EXPERIMENT SETUP

To numerically characterize the phase diagrams of the SSN, we simulated a minimal excitatory–inhibitory network with baseline parameters set to

$$J_{EE} = 2.5, \quad J_{IE} = 2.4, \quad J_{EI} = 1.3, \quad J_{II} = 1.0.$$

For each network configuration, the dynamics were iterated until convergence to a stable fixed point. External input strength was systematically varied over a wide range,

$$g \in \texttt{np.linspace}(0.5, 1000.0, 50000),$$

allowing us to probe both low-input and high-input regimes with fine resolution.

To construct two-parameter phase diagrams (e.g., $J_{EE}$ vs. $J_{II}$), we varied the selected pair of synaptic parameters over the interval $[0, 3]$ while holding the remaining parameters (e.g., $J_{EI}$ and $J_{IE}$) fixed at their baseline values. For each parameter combination and each input level $g$, the network was iterated to convergence, and the steady-state firing rate of the excitatory population was recorded. The input–output gain $dr_E/dg$ was then estimated using finite differences with respect to $g$. This procedure enables a systematic mapping of dynamical regimes, including nonlinear expansion and saturation, across the synaptic parameter space.

### B.2.2. FIG. 3 EXPERIMENT SETUP

To study the dynamical behavior of networks with multiple interacting subpopulations, we simulated a four-population SSN consisting of two excitatory (E1, E2) and two inhibitory (I1, I2) populations. We considered the most general symmetric configuration in which E1 and E2 are statistically identical, as are I1 and I2. Consequently, external stimuli were applied symmetrically to E1 and E2, ensuring that any differentiation in network responses emerged purely from recurrent dynamics rather than input asymmetries.

The recurrent connectivity matrix used in these simulations is given by

$$\mathbf{J} = \begin{pmatrix} 2.35 & 1.05 & 1.40 & 0.81 \\ 2.38 & 1.00 & 0.75 & 0.11 \\ 1.40 & 0.81 & 2.35 & 1.05 \\ 0.75 & 0.11 & 2.38 & 1.00 \end{pmatrix},$$

where each row specifies the incoming synaptic weights to a population (E1, I1, E2, I2), and each column denotes the presynaptic source.

For each simulation, the network dynamics were iterated until convergence to a stable fixed point. External input strength was varied over the range indicated in the corresponding figures. To isolate the role of lateral excitatory coupling between excitatory subpopulations, we additionally simulated a condition in which direct E–E interactions across populations were removed. Specifically, the cross-population excitatory connections $J_{E_1E_2}$ and $J_{E_2E_1}$ were set to zero, while all other synaptic parameters were left unchanged. This manipulation allows us to directly assess the contribution of inter-population excitation to the emergence of nonlinear amplification and dynamic coding behavior.

### B.2.3. GABOR ORIENTATION CLASSIFICATION EXPERIMENT SETUP

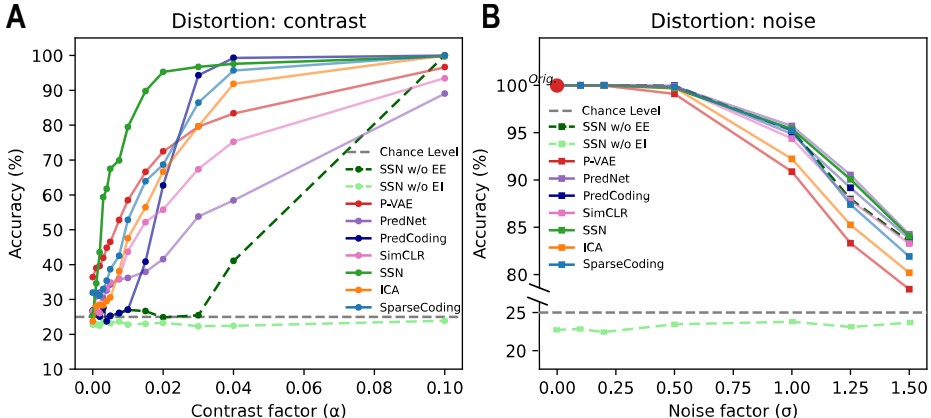

*Figure 7.* **Comparison with Degraded Inputs (Gabor grating orientations).** Gabor grating orientations classification under varying contrast and noise levels. Classification accuracy is shown for SSN, SSN w/o EE, SSN w/o EI, ICA, sparse coding, P-VAE, PredNet, predictive coding and SimCLR. The model size is limited to approximately 180 latent neurons for each.

**Data Preparation.** For model training, we randomly crop 9x9 patches from 10 natural scene images (512x512 in pixel) and construction a dataset of 128000 patches. For classifier training and evaluation, we generate a dataset of Gabor grating

patches, also sized 9x9 in pixel. Patches are varied in 4 different orientations ($0°$, $45°$, $90°$, $135°$), with a random angle fluctuation in the range of $-10°\text{~}10°$. We come up with 2 image distortion method, one is adjusting the contrast, the other is adding Gaussian noise. Each image $\mathbf{I}$, for contrast distortion, we have contrast factor $\boldsymbol{\alpha}$,

$$\mu = mean(\mathbf{I}) \tag{34}$$

$$\tilde{I} = \mu + \alpha(I - \mu) \tag{35}$$

$\alpha$ controls the extent of contrast adjustment, a 0 causes the image to be monochromatic, a 1 keeps the image unchanged.

For noise distortion, we have noise factor $\boldsymbol{\sigma}$,

$$\sigma_I = \text{std}(\mathbf{I}), \quad \epsilon_i \sim \mathcal{N}(0,1) \tag{36}$$

$$\tilde{I}_i = I_i + \boldsymbol{\sigma}\,\sigma_I\,\epsilon_i \tag{37}$$

0 means no noise is added, larger values make the image noiser. These distorted image patches are later used to test representation stability of the model.

**Training.** Sparse coding (Olshausen & Field, 1996) was trained directly on natural scene patches. The model contained 180 neurons and was optimized with an inference budget of 1000 steps and a convergence threshold of 0.01. The Independent Component Analysis (ICA) model (Bell & Sejnowski, 1997) was fitted to images whitened by Zero-phase Component Analysis (ZCA). All other models, including Poisson VAE (P-VAE) (Vafaii et al.), Predictive Neural Network (PredNet) (Rane et al., 2020), Predictive Coding (PredCoding) (Rao & Ballard, 1999), and SimCLR (Chen et al., 2020), were configured using the original hyperparameters reported in their respective papers, except that the dimensionality of the representation layer was standardized to approximately 180 units across models.

Our SSN was trained on the same natural image patches as the sparse coding model. It contained 180 excitatory neurons and 180 inhibitory neurons, using an inference budget of 200 steps. In addition, two ablation models were evaluated: SSN w/o EE, in which E↔E connections were removed, and SSN w/o EI, in which I→E connections were removed.

All models were pretrained on natural image patches. For downstream evaluation, a linear probe consisting of a single linear projection layer was trained on the inferred representations of distorted Gabor grating images. The classification labels corresponded to four grating orientations, encoded as 0, 1, 2, and 3. The learning rate and the number of training iterations were selected to ensure sufficient convergence of the classifier.

**Evaluation.** The classifier was used to predict the orientations of gabor grating images of each distortion type (contrast, noise). The prediction accuracy was evaluated across a range of distortion factor parameters for each model respectively, in 5 runs. As shown in Figure 7, the classifier of SSN outperformed other models in contrast distortion and achieved performance comparable to PredNet under noise distortion, thus suggesting that SSN provides more stable representations under degraded sensory inputs. In the ablation study, removing E↔E connections impaired the model's ability to support population coding under low-contrast conditions, resulting in degraded task performance. In contrast, removing I→E connections caused the model to fail on both tasks, because the absence of inhibitory feedback led to unstable neural representations.

### B.2.4. CIFAR10 CLASSIFICATION EXPERIMENT SETUP

**Data Preparation.** For model pretraining, CIFAR10 images were first converted to grayscale and then whitened. CIFAR-10 consists of 60,000 $32 \times 32$ color images from 10 classes, with 6,000 images per class. For classifier training and evaluation, we generated distorted images using two perturbation methods: contrast adjustment and additive Gaussian noise.

For contrast distortion, given an image $\mathbf{I}$ and a contrast factor $\alpha$, we first computed the standard deviation of pixel intensities:

$$\sigma_I = \text{std}(\mathbf{I}). \tag{38}$$

The distorted image was then obtained by normalizing the image and scaling it by $\alpha$:

$$\tilde{\mathbf{I}} = \alpha\frac{\mathbf{I}}{\sigma_I}. \tag{39}$$

Here, $\alpha$ controls the strength of contrast modulation. When $\alpha = 0$, image contrast is removed, whereas larger values of $\alpha$ increase the contrast.

For noise distortion, given a noise factor $\sigma$, Gaussian noise was added to the normalized image:

$$\sigma_I = \text{std}(\mathbf{I}), \quad \epsilon_i \sim \mathcal{N}(0, 1), \tag{40}$$

$$\tilde{I}_i = \frac{I_i}{\sigma_I} + \sigma \epsilon_i. \tag{41}$$

Here, $\sigma = 0$ corresponds to the clean normalized image, while larger values of $\sigma$ produce stronger noise corruption. These distorted images were used to evaluate the stability and robustness of the learned representations.

**Training.** The sparse coding model (Olshausen & Field, 1996) contained 14,400 neurons and was optimized with an inference budget of 1000 steps and a convergence threshold of 0.01. The Independent Component Analysis (ICA) model (Bell & Sejnowski, 1997) was fitted to images whitened by Zero-phase Component Analysis (ZCA). All other models, including Poisson VAE (P-VAE) (Vafaii et al.), Predictive Neural Network (PredNet) (Rane et al., 2020), Predictive Coding (PredCoding) (Rao & Ballard, 1999), and SimCLR (Chen et al., 2020), were configured using the original hyperparameters reported in their respective papers, except that the dimensionality of the representation layer was standardized to approximately 14,400 units across models. The SSN consisted of 14,400 excitatory neurons and 14,400 inhibitory neurons, and inference was performed for 200 steps. Each neuron received feedforward input through a $10 \times 10$ receptive field. Receptive-field locations were arranged with an overlap of 8 pixels between neighboring fields, yielding 144 spatial locations, with 100 neurons placed at each location. Neurons were distributed uniformly across the cortical sheet according to their receptive-field locations, with a small uniformly sampled spatial jitter added within each location. Recurrent connectivity was local: each neuron connected to its 180 nearest neighbors. For improved dynamical stability, we did not apply weight normalization to the $I \to E$ synapses.

For downstream evaluation, a linear probe consisting of a single linear projection layer was trained to predict class labels from the inferred representations of distorted CIFAR-10 images across a sweep of distortion parameters. The learning rate and number of training iterations were selected to ensure sufficient convergence of the classifier.

**Evaluation.** The classifier was then used to predict the labels of CIFAR-10 images of each distortion type (contrast, noise). The prediction accuracy was evaluated across a range of distortion parameters for each model respectively, in 5 runs. As shown in Figure 6, the classifier of SSN outperformed other models in low contrast condition, indicating its population coding capability. Also SSN representations suffered less from noise distortion than other models, thus indicating the superior stability of SSN.

### B.3. Properties of SSN trained in B.2.4

The additional analyses show that the large-scale SSN trained on CIFAR10 (Sec. B.2.4) develops adaptive coding strategies. Figure 8 illustrates contrast-dependent sharpening and the transition from population coding to sparse coding, whereas Fig. 9 presents the feedforward receptive fields learned by all 14,400 excitatory neurons.

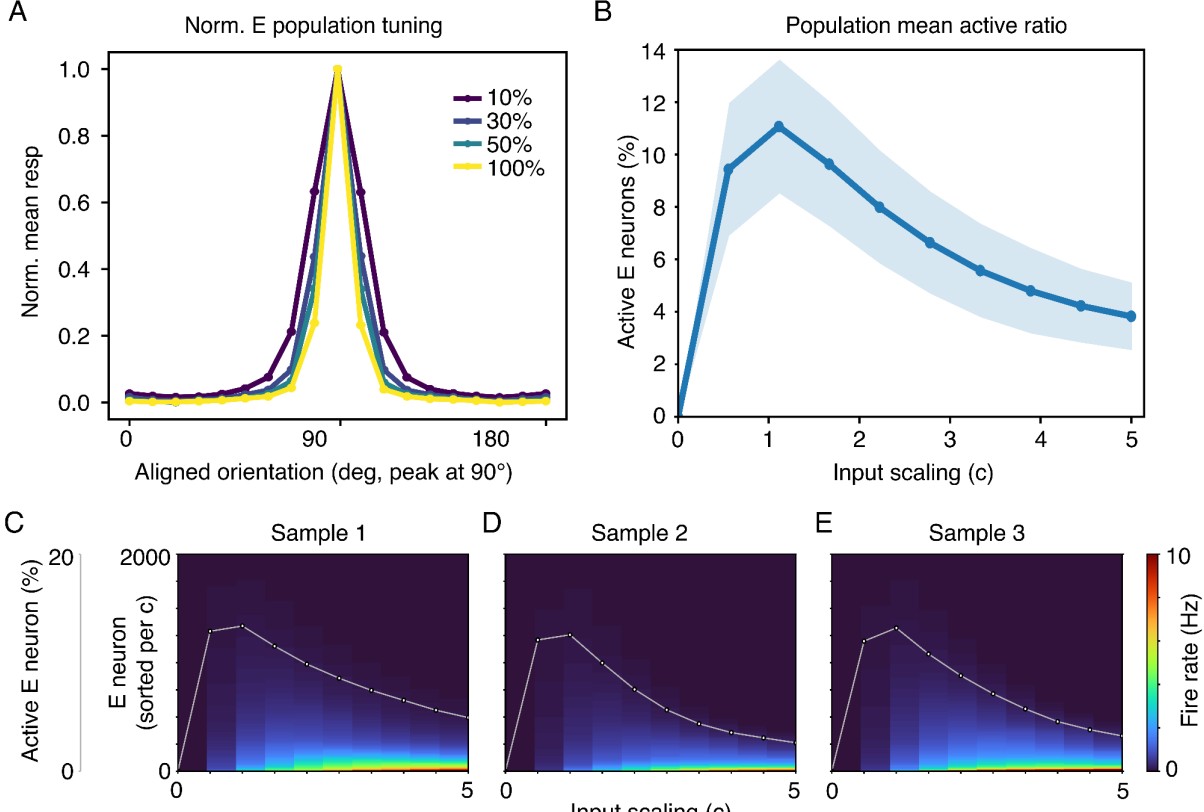

*Figure 8.* **The learned SSN spontaneously exhibits input-dependent dynamic coding strategies after training on CIFAR-10.** (A) As input contrast increases, the population tuning curve aligned at 90° becomes sharper, demonstrating contrast-dependent sharpening. (B) The coding regime shifts from population coding to sparse coding: with increasing contrast, the fraction of excitatory neurons participating in the representation decreases. (C–E) Three randomly chosen samples illustrate the activity patterns of the participating neurons across input scaling levels. Although the network contains 14,400 excitatory neurons, fewer than 2,000 neurons respond for each sample, showing that learned representations are highly sparse and input dependent.

Feedforward Receptive Fields of all 14400 Excitatory Neurons
（Receptive Filed: 10×10）

*Figure 9.* **Feedforward receptive fields of all 14,400 excitatory neurons.** Each patch shows the 10×10 feedforward receptive field of one neuron, and neurons are arranged according to their cortical positions in the large-scale simulation. Most neurons developed clear and structured receptive fields. Moreover, receptive fields at nearby cortical locations are not identical, indicating that the learned SSN forms diverse and effective sensory representations at scale.

