# OpenReview forum: "Adaptive Coding Emerges in Stabilized Supralinear Networks Trained with Local Plasticity"
_ICML.cc/2026/Conference — ICML 2026 regular_

### Official Review · Reviewer_5aaj · 2026-03-03

**Soundness:** 3
**Presentation:** 3
**Significance:** 2
**Originality:** 2
**Overall Recommendation:** 5
**Confidence:** 3

**Summary:**

This paper investigates the impact of excitatory and inhibitory lateral connections in recurrent neural network dynamics, mainly motivated by similar connectivity patterns in the brain. The authors focus on the stabilised supralinear network (SSN) as a model. Through a combination of theoretical and empirical results, the authors show that the dynamics of SSNs adaptively switch between two different regimes depending on the input statistics, revealing a cost-performance trade-off. In particular, the network dynamics shift from a population code with low-contrast or high-noise inputs to a sparse code for high-contrast, low-noise inputs. The authors also demonstrate how SSNs differs from and outperforms sparse coding and models based on independent component analysis.

**Compliance With Llm Reviewing Policy:**

Affirmed.

**Ethical Review Concerns:**

I recommend acceptance (5) because I think that, overall, the paper appears sound, clearly presented, while fair in originality and significance. My main concern was related to the framing and motivation of the paper from an ML/AI perspective, which the authors clarified and committed to toning down in the revised version. However, this work is in an area I am not an expert in and so my confidence score can only be 3 at best.

**Final Justification:**

Overall, I recommend acceptance (5) because, following the rebuttal, I believe the paper to be overall sound, clearly presented, while fair in originality and significance. My main concern was related to the motivation of the work from an AI/ML perspective, which the authors agreed to tone down in the revised version. However, I cannot provide a higher confidence score since the paper is in an area I am not very familiar with.

**Key Questions For Authors:**

Please see the major weaknesses above for my main questions and comments. I would also be curious to hear the authors' thoughts on the following:
* Why, in the authors' words, the gap in structure between modern AI and the brain has two main pathways? And why is the bottom-up, input-dependent pathway mainly driven by E-I recurrent networks?
* Could the authors comment on the relationship of their results with recent work (Schmutz et al., 2025) showing low- and high-dimensional dynamics can be reconciled under certain conditions in recurrent models?

**References**

Schmutz, V., Haydaroglu, A., Wang, S., Feng, Y., Carandini, M., & Harris, K. D. (2025). High-dimensional neuronal activity from low-dimensional latent dynamics: a solvable model. bioRxiv.

**Limitations:**

The authors briefly discuss the important limitations of the work in the conclusion.

**Strengths And Weaknesses:**

**Strengths**
* The paper is overall clearly written and structured.
* Clear motivation from a neuroscience perspective.
* Good synergy of theoretical and empirical results, building up from toy networks to models of increasing complexity.
* Detailed analyses clearly showing switching of dynamical regimes.
* Good comparison with standard methods including sparse coding and ICA.

**Weaknesses**

*Major*
* **Unclear ML/AI motivation**: While as the authors write AI models are far behind biological organisms in terms of adaptiveness and robustness, the link between explicit excitatory and inhibitory lateral connections appears to be too strong. Is there any evidence that such gaps in performance are due to such a difference in structure, as the authors argue? One can also think of many other differences in structure between modern artificial neural networks and the brain. Finally, mechanisms such as attention can arguably be seen as much more than normalisation given the complex interactions between queries, keys and values. For these reasons, I suggest that the authors tone down the ML/AI motivation.

*Minor*
* The authors provide good analyses of comparable neuroscience models such as sparse coding and ICA. However, an equally (if not more) influential theory of cortical function is predictive coding (e.g. see Rao and Ballard, 1999). Could the authors comment on the relationship or implications of their results (if any) with this model?
* The paper is full of typos, often having to do with double parentheses or missing spaces. See, for example, lines 50-1, 88-9, 95-6, 110-11, among others.
* In the caption of Figure 2, shouldn't $n$ be alpha? And what does the dotted line in Figure 2 indicate?
* The caption of Figure 3 could be improved to be more self-contained.

**References**

Rao, R. P., & Ballard, D. H. (1999). Predictive coding in the visual cortex: a functional interpretation of some extra-classical receptive-field effects. Nature neuroscience, 2(1), 79-87.

---

> ### Author Rebuttal · Authors · 2026-03-31
>
> We are truly grateful for your insightful comments and encouraged by your positive assessment of our work. We appreciate your suggestion to better calibrate the ML/AI motivation. In particular, we agree that our current framing may overemphasize the role of explicit E–I lateral recurrence. In the revised manuscript, we will reposition it as one of several plausible mechanisms that could help narrow the gap between biological and artificial systems in terms of robustness and adaptiveness, rather than as a primary or exclusive explanation. In addition, we did an ablation study to isolate the role of E-I recurrence in providing robustness and adaptiveness, please refer to the response for _reviewer W6YD_ and plots at https://tinyurl.com/rx13yu.
>
> **Predictive coding**
> We thank you for highlighting the importance of predictive coding, which we fully agree is a foundational framework for understanding V1 representations and cortical computation. In the original manuscript, our focus was on intra-cortical recurrent dynamics within a single layer, and thus predictive coding was not explicitly discussed. We appreciate this suggestion and will incorporate a discussion of predictive coding in the revised manuscript:
> Predictive coding posits a hierarchical architecture in which higher areas send predictions and lower areas transmit residual errors, successfully accounting for extra-classical receptive field effects under natural image statistics. Our work is distinct in both mechanism and scope: rather than a hierarchical predictive framework, we study trainable within-layer E–I recurrent circuits with local plasticity, and show that such circuits can self-organize an input-dependent transition between population and sparse coding regimes. Importantly, we view these two perspectives as complementary rather than competing. As discussed in the predictive coding literature, lateral recurrent interactions are not excluded and may play a role in local error computations. From this perspective, the robustness observed in our model can be interpreted as arising from coordinated activity that effectively minimizing the total prediction errors within the layer. This suggests a natural integration of our framework with predictive coding as a promising direction for future research. We thank the reviewer for this insightful suggestion.
>
> Motivated by your suggestion, we included predictive coding models in our revised experiments —- the classical formulation (Rao & Ballard, 1999) and a modern variant PredNet (Lotter et al. 2017) -— and evaluated them under the same settings with CIFAR-10. Plots are in https://tinyurl.com/rx13yu.
>
> |CIFAR-10 condition|c=0.073|0.430|1.250|1.750|σ=1.33|2.67|5.33|
> |-|-|-|-|-|-|-|-|
> |SSN|**14.31**|**43.48**|**48.18**|**47.77**|**43.91**|**34.59**|**16.90**|
> |PredNet|10.00|10.91|31.38|46.19|31.11|19.73|14.20|
> |PredictiveCoding|10.59|17.29|31.42|32.52|26.75|17.85|12.64|
>
> **Schmutz et al.** is a highly insightful study. It investigates whether high-D observed cortical activity reflects low- or high-D latent dynamics under different input categories and spontaneous activity, whereas our work studies how explicit E–I lateral circuitry in a trainable SSN reshapes coding regimes as input changes. Their results are therefore complementary to us rather than overlapping. In particular, their finding that low-D pre-activations can still generate high-dimensional post-activation responses cautions against equating low latent dimension with sparse activity; our work instead focuses on neuronal activity from the perspective of input-dependent redistribution and robustness, thereby offering a complementary viewpoint. Exploring how these two perspectives can be integrated represents a promising direction for future work.
> _We will include this paper in the revised manuscript._
>
> **Two pathways**: The top-down and bottom-up pathways are used here to help conceptually dissect how signal processes flow in the brain (in modern AI as well) and set the stage for E-I recurrence's role in the latter pathway, which is indeed one of the biggest structural differences between brain and AI, but we are not trying to attribute the two pathways directly to the gap between brain and AI.
> The importance of E-I recurrence has been established by the experiments that find the feed-forward signal substantially amplified by the recipient layers in the cortex (with preferred stimuli, the cortex contributes almost 2/3 of the total excitation, [Lien & Scanziani 2013](https://www.nature.com/articles/nn.3488)) and this recurrence plays an important role ([Pattadkal et al. 2024](<https://www.cell.com/neuron/fulltext/S0896-6273(23)00880-2>)). Finally, We are sorry that the text around "two pathways"" leads to this ambiguity, we will clarify it in the revised manuscript.
>
> **Minors**: We regret the obvious typos and mistakes spread across the text and figures, we will carefully examine the manuscript and rectify these errors in the revised one.

---

> > ### Author Rebuttal · Reviewer_5aaj · 2026-04-01
> >
> > I thank the authors for the response and additional experiments including informative ablations. I selected **(a)** because the rebuttal addresses all of my concerns. As agreed by the authors, the revised version of the manuscript should tone down the ML motivation, in line with feedback from other reviewers.

---

> > > ### Author Response · Authors · 2026-04-01
> > >
> > > Thank you very much for your time and effort in reviewing our manuscript, as well as for your positive assessment, which is truly encouraging to us. We are also very glad that our rebuttal has addressed your concerns.
> > >
> > > Your comments are highly insightful and constructive. We will revise the manuscript accordingly, including refining the ML/AI motivation as suggested. We will also incorporate the additional experiments from the rebuttal into the main text.
> > >
> > > Thank you again for your valuable feedback.

---

### Official Review · Reviewer_W6YD · 2026-03-06

**Soundness:** 2
**Presentation:** 3
**Significance:** 2
**Originality:** 2
**Overall Recommendation:** 3
**Confidence:** 3

**Summary:**

The paper studied the importance of lateral connections in stabilized supralinear networks (SSNs) with explicit excitatory–inhibitory during unsupervised visual representation learning. The authors first provided a theoretical analysis of a minimal two-population SSN, identifying a nonlinear expansion regime and an activity redistribution mechanism. Then they trained a larger-scale SSN using local Hebbian plasticity rules on whitened natural image patches. The trained network develops Gabor-like receptive fields and shows contrast-dependent activity patterns. The authors report that the model transitions from population coding under low contrast to sparse coding under high contrast and achieves better robustness than ICA and sparse coding under degraded inputs.

**Compliance With Llm Reviewing Policy:**

Affirmed.

**Key Questions For Authors:**

1. Can the authors provide an ablation study removing E–I lateral connections to verify that the reported improvements are specifically caused by this mechanism?
2. What objective or principle (if any) is implicitly optimized by the BCM and correlation-based learning rules used in training?
3. Have the authors verified that the learned weights of the trained SSN fall within the nonlinear expansion regime predicted by their theoretical analysis (Eq. 7, Fig. 2D)? Demonstrating this connection would significantly strengthen the paper’s coherence between theory and experiments.
4. Is there more evidence to justify that the proposed SSM exhibits shared underlying dynamics with real neural system?

**Limitations:**

Yes.

**Strengths And Weaknesses:**

Soundness:
The technical support for the central claims is limited.
1. The theoretical analysis in Section 3 focuses on simplified two-population systems and does not clearly demonstrate that the same mechanisms hold in the larger trained network, which is not sufficient to support the succeeding Section 4.
2. Several arguments appear heuristic rather than rigorously justified. For example, the claim that biological systems require supralinear responses because inputs are not normalized is not clearly derived from the analysis.
3. Important experimental controls are also missing. The paper attributes improvements to explicit E–I lateral connections, but no ablation study (e.g., an SSN without these connections) is provided. Without such baselines, it is difficult to determine whether the observed behavior is actually caused by the proposed mechanism.
4. The learning process in Section 4 is difficult to interpret because the Hebbian training rules do not correspond to a clearly defined optimization objective and are only an approximation of the real synaptic rules of neurons. The author did not provide any comparison with real neural activity data, except for the brief statement that the tuning profile is consistent with a 2002 study in section 4.2.

Presentation:
The overall structure of the paper is reasonable, but several parts of the presentation lack clarity. Some terminology and notation are insufficiently defined. For example, the term “admissible region of state space” is unusual in this context and not clearly explained. Equation (3) introduces the operator `det(J)` without explicitly defining the connectivity matrix. In addition, certain figures are difficult to interpret: for instance, Figure 2D does not clearly explain the meaning of the color intensity. The reasoning in some sections could also be more carefully presented, particularly where theoretical intuition is used to motivate model properties.

Significance:
The question about what functional role do explicit E-I lateral connections play in sensory processing is important, and the proposal that such connections enable adaptive switching between population coding and sparse coding is a good conceptual contribution. However, the extremely limited evaluation scope severely limits the demonstrated significance. The model operates on 9×9 image patches with only 360 neurons; the only task is 4-way orientation classification. There is no evaluation on any standard vision benchmark, and the proposed SSN model is also not fully justified to be consistent with real neural system. The paper’s ambitious framing is not matched by the current experimental evidence.

Originality:
SSNs themselves are not new, and the individual components (E-I connections, BCM rule, CM rule, Gabor-like RF emergence) have been studied extensively before. The main contribution appears to be the combination of these ideas and the interpretation of coding regime transitions, rather than the introduction of fundamentally new theoretical or methodological advances.

---

> ### Author Rebuttal · Authors · 2026-03-31
>
> We sincerely thank you for your careful and constructive assessment. We appreciate the recognition of the importance of our work in understanding V1 coding strategies in biological systems, particularly the functional role of E–I lateral connections, as well as the positive evaluation of our conceptual contribution on adaptive coding strategies.
>
> **Standard benchmark with ablation**
> Following your valuable suggestion, we conducted ablation to isolate the role of E–I lateral connectivity. Specifically, we removed (i) E<->E connections which leads to an absence of nonlinear expansion at low contrast, (ii) I->E connections which leads to runaway excitation to compare with the full SSN under the same setting, plots are in https://tinyurl.com/rx13yu.
>
> |condition|full|w/o EE|w/o IE|
> |-|-|-|-|
> |c=0.001|0.35|0.27|0.23|
> |c=0.04|0.98|0.41|0.22|
> |c=0.1|1.00|1.00|0.24|
> |σ=0|1.00|1.00|0.23|
> |σ=1.5|0.84|0.84|0.24|
>
> To make our claims solid, we trained and re-evaluated a larger-scale version of our model (120×120×2 E/I units) using CIFAR-10 against various models (based on you and other reviewers' comments) under identical size constraints and evaluation protocols. The larger-scale model reproduces all properties reported in the main text (plots: https://tinyurl.com/g65poz) and shows superior performance in robustness, please refer to the response to _reviewer Brpz_ and https://tinyurl.com/rx13yu for plots.
>
> **Vs. real neural system**
> Though well motivated, we agree that our framing is too ambitious against the scarce direct evidence available due to the challenges of measuring large-scale activity simultaneously, however there still exists some indirect evidences in V1:
> - Contrast-dependent spatial integration (Sceniak et al. 1999 Nat Neurosci.):  Neurons have a larger size tuning (through lateral E recruitment) at low contrast, while more suppression and smaller preferred sizes at high contrast.
> - Contrast-dependent sharpening of orientation tuning (for mouse, ferret and monkey): we have added plots at https://tinyurl.com/8askd1 to compare tuning curves and scatter plot of O/P ratio with figures in Li et al. 2012 and Johnson et al. 2008.
> Both suggest a transition from population to sparse coding as contrast increases.
>
> **Learning rules**
> BCM rule is used on feed-forward connections to both E and I, and can be seen as a approximation of a triplet STDP variant found in the visual cortex (Zenke et al. 2013 PloS Comp Biol.) which has been shown to promote super-Gaussian representations and sparsity, thereby sufficient for the emergence of RFs (Brito & Gerstner 2016 PloS Comp Biol.). The CM rule drives the connection weights symmetrically based on pair-wise correlation thus it implicitly optimized for stronger connection between E neurons that have similar RFs, supporting the broader recruitment when input signal is weak. In addition, the E->I->E pathway ensures that no pair of E can have exactly the same RF, so sparse coding can still be achieved through WTA when input signal is strong.
>
> **Originality**
> We agree that individual components used in our framework are established previously. However, our contribution is the identification of a mechanism by which they interact to produce emergent coding behavior that is not present in any individual component alone, which to our knowledge, has not been reported in any prior studies. (For detailed elaboration and a better positioning of our contribution please kindly check the response for _reviewer Brpz and 8mru_).
>
> **Bridge theory-to-experiment**
> We thank you for the constructive comment and agree that verifying whether the learned SSN operates in the regime predicted by Eq. 7 is critical for connecting theory to experiments.
> To verify this, we averaged the effective coupling strengths estimated by summing connections from neurons with similar preferred orientations (< 20°), thereby with correlated activity and most relevant for the nonlinear expansion: JEE=0.834, JEI=0.455, JIE=0.760, JII=0.271.
> Using these values, we recomputed the phase diagram in Fig.2D with Eq.7 and marked the operating point of the learned network which indeed falls in the nonlinear expansion regime (https://tinyurl.com/8uzj1d).
>
> **Presentation**
> We sincerely thank you for your careful reading. We will revise the manuscript to improve clarity and reasoning as suggested, including the changes below:
> - define J as the connectivity matrix $ [J_{EE}, -J_{EI}; J_{IE}, -J_{II} ] $ before Eq. 3.
> - include colorbars in Fig.2D (check https://tinyurl.com/ha712a).
> - change "admissible region of the state space" to "set of feasible states of activity".
> - make it explicit that the proposal that implementing a SSN-like nonlinear gain could be how biological systems adapt to changes in input conditions in the absence of strict normalization pre-process required for sparse coding and ICA is based on experimental findings and heuristics. The preceding analysis merely supports its sufficiency not necessity.

---

> > ### Author Rebuttal · Reviewer_W6YD · 2026-04-02
> >
> > Thanks for addressing my concerns. I have no further questions.

---

> > > ### Author Response · Authors · 2026-04-02
> > >
> > > Thank you again for your time and thoughtful evaluation of our work. We are glad that our rebuttal has addressed your concerns.
> > >
> > > We hope that the clarifications and additional results provided in the rebuttal help to further strengthen the presentation and support of our contributions.
> > >
> > > Thank you once again for your insightful and constructive feedback.

---

### Official Review · Reviewer_Brpz · 2026-03-12

**Soundness:** 2
**Presentation:** 3
**Significance:** 2
**Originality:** 2
**Overall Recommendation:** 2
**Confidence:** 3

**Summary:**

This research paper put forward a viewpoint that the Stabilized Supralinear Networks which are short for SSNs, have clear excitatory-inhibitory lateral connections that is usually called E-I connections, and this kind of network can finish dynamic switch between two coding modes: one is population coding, this mode will work when the input signal is low contrast or degraded condition, the other is sparse coding, this mode will work when the input signal is high contrast condition.
The researchers of this paper do theoretical analysis aiming at the nonlinear expansion regime that exists in the SSN with two populations, and they also do relevant proof test: use the Hebbian-like local learning rules to train the recurrent E-I network, and take the natural image patches as training data, after the training process, this network can get the receptive fields that look like Gabor type, and also realize the contrast-dependent sharpening function successfully.
The authors of this manuscript think that this kind of dynamic coding strategy can make a good balance between the metabolic cost, computational cost and the working performance of the model. Besides, when dealing with the signal representation that has noise problem and different contrast levels, this model has better effect than the classical Sparse Coding model and the Independent Component Analysis model which is short for ICA model.

**Compliance With Llm Reviewing Policy:**

Affirmed.

**Final Justification:**

The rebuttal improves the paper by adding stronger experimental validation and clarifying the intended contribution, which partially addresses my concerns. However, issues regarding novelty framing and the cost claim remain only partly resolved, so I maintain my original assessment.

**Key Questions For Authors:**

1. Could you make a clear and separate explanation about the theoretical dynamics content in Section 3: which parts are the new and original contributions of this paper, and which parts are the already known and established basic properties of SSNs that have been proved in the previous published research literature?
2. Why the modern unsupervised sparse network structures are not chosen and added into the baseline comparison experiments?
3. In which practical and real machine learning application situations, this kind of dynamic switching function of the network can really reduce the calculation amount or memory occupation? Are there any related hardware simulation experiments to support the claim about "cost reduction" put forward in this paper?
4. The authors said their methods are able to train large model. What are the main and key obstacles and bottleneck problems that stop the authors from training this SSN model on the real large-scale images (such as 224×224 size images), instead of only using the small 9×9 size image patches to do the training test?

**Limitations:**

The model is very small. The dataset is very small. The evidence of energy saving is very weak. The authors should show these in the limitation.

**Strengths And Weaknesses:**

## Strengths:

This paper has made a good mathematical deduction work in Appendix A part, it clearly gives the boundary conditions about the superlinear gain that exists in the two-population Stabilized Supralinear Networks, and this mathematical derivation process is concise and elegant, without too much redundant content.

The explanation about the biological rationality of this model is very sufficient and persuasive, especially the key proof part: using BCM rule and CM rule these two kinds of local learning rules, can make the SSNs gradually form the receptive fields that are similar to the real primary visual cortex V1, and also can show the sharpening function that changes with the contrast level, this part has strong convincing power.

Besides, this paper connects the input contrast level with the automatic switching function of different coding modes, which are population coding and sparse coding, this kind of research viewpoint is novel and meaningful for the study of representation learning, it brings a new thinking direction for this field.

## Weaknesses:

Overstated Novelty and Missing Context: The nonlinear expansion and inhibition-stabilized regimes are literally the defining features of SSNs. Framing the high-contrast suppression of weak units as a "novel dynamic coding strategy" borders on repackaging known SSN behaviors. The theoretical daylight between this work and prior SSN analyses needs to be clarified.

Inadequate and Outdated Baselines: The experimental evaluation (Section 4.3) benchmarks the proposed model against ICA (1997) and classical Sparse Coding (1996). Evaluating against 30-year-old algorithms is insufficient for an ICML submission in 2026. The authors must compare against modern unsupervised representation learning methods, predictive coding networks, or modern sparsity-inducing autoencoders.

Unverified Claims on Computational Cost: The title and abstract heavily emphasize "balancing cost and performance." However, "cost" is measured solely as biological metabolic cost (number of firing neurons). For a machine learning venue, this should be contextualized with actual computational metrics (e.g., FLOPs, memory footprint, wall-clock time, or simulated energy consumption on neuromorphic hardware). Without this, the cost-saving claims are practically unsubstantiated.

Misleading "Large-Scale" Claims: The authors state on Line 78 that they trained a "large-scale SSN". Section 4.1 reveals the model processes 9×9 whitened patches using only 360 total neurons. This is a toy model. And no scaling to standard datasets (e.g., CIFAR, ImageNet) severely weakens the paper's impact.

---

> ### Author Rebuttal · Authors · 2026-03-31
>
> We sincerely thank you for the time and effort you dedicated to reviewing our paper and for recognizing the relevance of our work to understanding visual representations of biological intelligence. We understand that your main concerns, particularly Q2 and Q4, the simplicity of the experimental design and the limited scale of the trained SSN. In the original manuscript, our experiments were primarily designed to explain biological visual phenomena, thus only focused on comparisons with canonical models in visual neuroscience, i.e., sparse coding and ICA, which provide normative accounts of V1 responses.
>
> In direct response to your concerns, we have conducted additional experiments. Specifically, we trained a larger-scale SSN (120×120×2 E/I units) on CIFAR-10. The resulted network reproduces all key properties reported in the main text, including RF formation, contrast-dependent sharpening, and the transition between population and sparse coding regimes (see https://tinyurl.com/g65poz). We further extended our evaluation to include modern unsupervised models: SimCLR, two predictive coding models (Rao & Ballard 1999; Lotter et al. 2017), and a Poisson autoencoder (Vafaii et al. 2024). For fairness, all models were implemented with a single layer (14400 excitatory units), and evaluated with a linear readout, the exactly same configuration as our SSN.
>
> |model/condition|ɑ=0.073|0.430|1.250|1.750|σ=1.333|2.667|5.333|
> |-|-|-|-|-|-|-|-|
> |SSN|**14.31**|**43.48**|**48.18**|47.77|**43.91**|**34.59**|**16.90**|
> |PVAE|13.55|20.37|36.17|37.58|34.15|24.79|14.84|
> |PredNet|10.00|10.91|31.38|46.19|31.11|19.73|14.20|
> |PredictiveCoding|10.59|17.29|31.42|32.52|26.75|17.85|12.64|
> |SparseCoding|11.26|32.37|35.60|35.11|27.03|19.20|13.32|
> |ICA|11.91|22.87|26.48|26.31|11.96|10.68|10.64|
> |SimCLR|10.00|11.73|34.92|**54.89**|37.80|20.02|12.76|
>
> Across both the original toy setting and CIFAR-10, our SSN consistently demonstrates superior robustness under low-contrast and high-noise conditions, through its adaptive coding. While SimCLR achieves higher accuracy at high contrast, it is expected given its objective and lack of sparsity; notably, it utilizes substantially denser representations (40% active units) compared to the SSN (8%) at high contrast. Please check https://tinyurl.com/rx13yu for complete results.
>
> **Novelty**
> We thank you for giving us the opportunity to clarify the distinction between our contributions and established SSN properties. Our central claim is not that inhibition-stabilized or supralinear amplification regimes are new, but that they can be _learned_ with a fully local and biologically plausible framework and the learned SSN offers the emergence of adaptive coding strategies. In contrast to prior SSN studies that rely on hand-crafted connectivity or specialized training procedures, we show that BCM/CM-based local plasticity is sufficient for SSNs to learn V1-like receptive fields and, *crucially*, to exhibit an input-dependent transition between population and sparse coding regimes. In addition, we demonstrate the emergence of contrast-dependent sharpening in the learned SSN. To the best of our knowledge, these contributions are new (see also response to _reviewer 8mru_ Positioning part).
>
> **Metabolic vs. Computational Cost**
> Despite being mainly a _computational neuroscience_ paper, we do agree with you that our notion of “cost” primarily reflects metabolic cost, using the number of active neurons as an energy proxy based on experimental evidence. In the revised manuscript, we will be explicit about the cost being merely biological and one need to fit a real energy function for cost estimation, and avoid conflating it with computational cost.
> Our primary goal is to elucidate the circuit mechanisms underlying adaptive coding strategies, rather than addressing computational efficiency in modern hardware. Nevertheless, we thank you for pressing on finding where our work can contribute in real world applications. In neuromorphic computing, the major bottleneck is associated with the large-scale communication between neurons through spikes, leading to substantial computational costs. Although hardware like "multicast" has been introduced to support the delivery of a single spike to multiple cores (neurons) at the same time and alleviates this constriction to some extent (Navaridas et al. 2015 Parallel Compt.). For distributed low-rate firing across many neurons, the system is still bottlenecked by the effective multicast capacity of the hardware: routing entries, replication points, hot-link bandwidth, and buffering. Our work provides an analogous understanding of this trade-off situation, and supports the transition to sparse coding when the input conditions allow for it.
>
> We thank your for all the valuable suggestions, they have helped place our claims on firmer empirical footing. We will incorporate them into the revised manuscript.

---

> > ### Author Rebuttal · Reviewer_Brpz · 2026-04-03
> >
> > Thank you for your rebuttal. My concerns have been partly solved.
> > I appreciate that the authors have added more sufficient experimental evidence to answer my previous concerns. Especially, adding more modern baseline methods and larger-scale experiments greatly improve the experimental part. This well solves my main concern that the original evaluation only used old comparisons and small-scale experiments.
> >
> > My main remaining problem is about the novelty and cost claim.
> > I also appreciate the explanation about novelty. The rebuttal makes it clear that the inhibition-stabilized / supralinear SSN regimes themselves are not new. The real contribution is that these behaviors can be learned by local and biologically plausible plasticity rules, and can achieve adaptive switching between coding regimes. This explanation is very helpful. However, I still believe that the author's paper presentation is misleading. And this part is more like a finding. So I suggest the authors clearly emphasize this difference in the revised paper.
> > The rebuttal has explained that cost mainly refers to biological/metabolic cost instead of computational cost. However, this makes the current expression “balancing cost and performance” too strong for a machine learning conference, unless it is strictly limited.
> > The rebuttal still lacks direct evidence such as FLOPs, memory, running time, hardware or neuromorphic simulation results. Therefore, I think this problem is only partly solved. I suggest the authors modify the title and abstract, and clearly point out this limitation.
> > Overall, the rebuttal has improved the paper and resolved parts of my previous concerns, but not all completely.

---

> > > ### Author Response · Authors · 2026-04-08
> > >
> > > Thank you for the time and effort that you have devoted to reviewing our paper. We are glad that the additional experiments have helped strengthen our work and addressed part of your concerns. We also greatly appreciate your thoughtful feedback regarding the novelty and statements on "cost". Accordingly, we will ensure a precise and well-calibrated presentation in the revised manuscript.
> > > ## Novelty
> > > We are glad that the clarification of our contribution and positioning in the rebuttal has been helpful. We also agree that presenting our contributions together with established SSN properties in the original manuscript may be misleading. We will revise for a clearer differentiation:
> > >
> > > In **Contributions**:
> > >
> > > - A biologically plausible learning framework for the established SSNs.
> > > - The emergence of an input-dependent coding strategy (including contrast-dependent sharpening) from learning.
> > > - Empirical evidence that this adaptive strategy improves robustness under degraded inputs vs. alternative methods.
> > >
> > > We will add the paragraph from our response to _reviewer 8mru_ in **Related Works** section to clearly position our work within the SSN literature.
> > >
> > > We agree that the observed adaptive coding strategy in our SSN model is best characterized as a finding, as it emerges from the learning dynamics rather than some explicit designs. At the same time, we would like to clarify why we believe this finding is non-trivial and scientifically meaningful.
> > >
> > > Understanding how local connectivity affects information processing in the visual cortex is a central topic in neuroscience. Although it is well-known that V1 exhibits strong recurrence for both E and I neurons, most models use feed-forward or inhibitiory recurrence only networks (ICA, sparse coding and predictive coding) for encoding features, while the functional role of strong E-E recurrence are less understood.
> > >
> > > Motivated by this gap and the adaptiveness of visual processing in the brain, we trained a SSN model with biologically plausible plasticity using natural images where an input-dependent transition between population coding and sparse coding emerges. To our knowledge, while SSN has been well studied, it is known to be hard to train, and the emergence of such adaptive coding behavior from a unified, biologically plausible learning process has not been reported. We further demonstrated the trained SSN's superior robustness and adaptiveness empirically. This adaptiveness also manifests as contrast-dependent sharpening of orientation in our model, consistent with observation in mouse, ferret, and macaque V1, for which our work offers a functional explanation through learning.
> > >
> > > In summary, our finding provides a mechanistic account of how E–I recurrence can support adaptive coding in V1 and may shed lights on designing AI with brain-inspired adaptiveness.
> > >
> > > ## Cost
> > >
> > > We fully agree that, in the context of ML, “cost” is typically interpreted as computational cost (e.g., FLOPs or memory), rather than metabolic cost. We acknowledge that our original phrasing (“balancing cost and performance”) may therefore be misleading, and will clarify that “cost” in our work refers to metabolic cost, proxied by the number of active neurons. We do not claim computational cost-effectiveness on conventional hardware and will ensure the revised manuscript makes no such implication.
> > >
> > > Following your suggestion, we will revise the manuscript to avoid the misunderstanding of "cost" and overstatements about "balancing cost and performance" in our work:
> > >
> > > **Title**
> > >
> > > > Adaptive Coding Emerges in Stabilized Supralinear Networks Trained with Local Plasticity
> > >
> > > **Abstract (excerpt)**
> > >
> > > > During the transition, the network shifts from population coding, which extracts features from low-contrast or noisy inputs by recruiting more neurons, to sparse coding at high contrast, which encodes inputs using smaller sets of neurons. This reduction in the number of active neurons is generally associated with lower metabolic demand as suggested by models and experiments (Yi et al., 2016; Hasenstaub et al., 2010).
> > >
> > > **Limitations**
> > >
> > > > (line 429)...spike frequency adaptation, _affecting energy functions for calculating the exact metabolic cost. Here, we simplify it by using the number of active neurons as a proxy (Yi et al., 2016; Hasenstaub et al., 2010), which does not translate to computational cost on conventional hardware. That said, whether the observed adaptive coding strategy could yield practical benefits in neuromorphic systems (Navaridas et al. 2015) remains to be investigated in future work._ Empirically... (line 431)
> > >
> > > Misc:
> > >
> > > - Change "cost-effective" to "metabolically cost-effective" in line 091, 120, 381, 410 and 428.
> > > - Remove "...to balance cost and performance" from the sentence at line 418 in **Conclusion**.
> > > - Remove claims about "cost" in **Contribution** (see the reply under **Novelty**).
> > >
> > > We thank you again for your constructive feedback, and hope that the revisions help address your concerns.

---

### Official Review · Reviewer_8mru · 2026-03-12

**Soundness:** 4
**Presentation:** 3
**Significance:** 3
**Originality:** 3
**Overall Recommendation:** 5
**Confidence:** 3

**Summary:**

The authors study stabilized supralinear networks as a model of early visual cortex. They focus, in particular, on how excitatory and inhbitory lateral connections drive more distributed representations at lower contrasts and more winner-take-all dynamics at higher contrasts. They further train a SSN on natural image data using local synaptic learning rules and find that the theoretical intuitions they previously developed in simple settings apply to this network as well. In particular, this explains contrast-dependent sharpening effects also observed in mouse visual cortex. Finally, they show that the representations arising from the SSN yield improved classifier performance compared to those of alternative models of V1.

**Compliance With Llm Reviewing Policy:**

Affirmed.

**Final Justification:**

I continue to think this paper would be a good addition to ICML.

**Key Questions For Authors:**

1. Can you further clarify how your insights here relate to previous studies of SSNs? For example, the related work discussion leaves unclear whether contrast-dependent sharpening was previously observed in this models. (As far as I understand not, in which case it would be nice to emphasize this.) More broadly, clarifying which of the observations you're making here are novel would help the reader more easily understand the specific contributions.

**Limitations:**

Yes

**Strengths And Weaknesses:**

This is a well-written paper that provides a valuable intuition which is supported by larger-scale simulations as well. The figures are well-designed and provide a useful conceptual overview. I also really appreciated the multi-headed evaluation: starting out with a simple analytical intuition, followed by a model that is trained on naturalistic stimuli, and finally analyzing the potential usefulness of these representations. The authors' provided method that allows them to train SSNs on high-dimensional stimuli may also be useful for future work. Finally, I appreciated the careful discussion of limitations of the work that may be a useful inspiration for future work as well. Overall, I would therefore recommend acceptance to ICML.

My concerns about this submission are relatively minor. Primarily, I think the authors should discuss in more detail which of the observations made here (if any) were known from previous SSN studies. On the presentation end, I also think adding a figure of physiological studies showcasing e.g. contrast-dependent sharpening might be useful in helping the reader evaluate these comparisons.

A couple additional minor notes:

- L. 29: I think the statement that this gap in performance is implied is a bit strong and I would suggest weakening the statement a little bit (along the lines of "The first gap in performance could potentially be addressed by closing the second gap")
- L. 206: do you mean superlinear and sublinear?
- L. 221: can you explain in more detail how this is consistent with experimental observation?
- Evaluating the extent to which the connectivity structure has diagonal bands in Fig. 4D qualitatively is a bit difficult. Is it possible for you to provide a more quantitative evaluation?

---

> ### Author Rebuttal · Authors · 2026-03-31
>
> We sincerely thank you for your time and effort devoted to carefully reading our manuscript. We greatly appreciate your valuable feedback, as well as your positive assessment and constructive suggestions, which will significantly help improve the clarity, presentation, and overall quality of our work.
>
> **Positioning.** Thanks for you suggestion on a better positioning by explicitly distinguishing established SSN properties from our specific contribution. We will add the following dicussion to the revised manuscript:
>
> Foundational SSN work established the core dynamical regime where supralinear responses stabilized by inhibition, and used this framework to explain normalization, surround suppression, and other contrast-dependent effects, but not contrast-dependent sharpening of orientation tuning ([Ahmadian et al. 2013](https://doi.org/10.1162/NECO_a_00472); [Rubin et al. 2015](https://doi.org/10.1016/j.neuron.2014.12.026)). Later SSN extensions focused on quenching variability ([Hennequin et al. 2018](https://doi.org/10.1016/j.neuron.2018.04.017)), cell-type specificity ([Millman et al. 2020](https://doi.org/10.7554/eLife.55130)), attention ([Lindsay et al. 2020](https://doi.org/10.1101/2019.12.13.875534)), bistable, oscillatory and persistent regimes ([Kraynyukova and Tchumatchenko 2018](https://doi.org/10.1073/pnas.1700080115)), and spatial/contextual effects ([Obeid and Miller 2025](https://doi.org/10.1523/ENEURO.0459-24.2025), [Wu et al. 2026](https://doi.org/10.64898/2026.02.06.704473)), and contrast-dependent gamma oscillation ([Holt et al. 2024](https://doi.org/10.1371/journal.pcbi.1012190)) rather than the sharpening effect (although Rubin et al., 2015 and Hennequin et al., 2018 observed contrast invariance in SSN). Thus, our claim is not that inhibitory stabilization, supralinear amplification, or contrast-dependent gain control are new SSN properties; rather, our novelties are (i) a SSN trained with local plasticity, learned from natural images, can develop V1-like receptive fields and spontaneously exhibit both a contrast-dependent transition from broader population coding at low contrast to sparser coding at high contrast, (ii) it has the same dynamical regime where SSN can achieve contrast-dependent sharpening of orientation tuning as experimentally observed (the classical contrast invariance observed in cat is not true in mouse, ferret, and monkey). To the best of our knowledge, this sharpening effect was not previously reported in SSN, and we will revise the related-work section to make this distinction explicit.
>
> **Presentation**: We agree that we need to strengthen the physiological and quantitative presentation accordingly. We will add plots for the comparison with contrast-dependent sharpening from experiments in macaque and mouse V1 (Johnson et al., 2008, Li et al. 2012) in the revised manuscript (preview plots at https://tinyurl.com/8askd1), and they do match qualitatively well.
>
> **Minors**:
> - _L. 29_: We will _tune down_ the statement about "the gap in performance being implied by the gap in structure" to structure being one of the potential causes, due to the lack of decisive evidence in experiments. We will replace it with "The first gap in performance could potentially be addressed by closing the second gap in structure, which is one of the possible causes", built on the well-versed sentence that you suggested.
> - _L. 206_: Yes, it should be supralinear instead of just nonlinear.
> - _L. 221_: We want to thank you for spotting this mistake! We put the wrong reference here, it should be [Chettih & Harvey 2019](https://www.nature.com/articles/s41586-019-0997-6) which supports stronger suppression for similar preference, we will update the text and figure/caption with better presentation, i.e., cross-preference suppression should be evaluated with $(E_{1}(c_{1},0) - E_{1}(c_{1}, c_{2}))$ instead of $E_{1} - E_{2}$ (see updated https://tinyurl.com/6uzyj1).
> - _Quantification_: We provide a quantitative analysis of the diagonal structure in Fig. 4D (see https://tinyurl.com/a1x0zk, to be included in the revised manuscript). Specifically, we measure the mean synaptic weight as a function of feature difference (Δθ, Δφ) and fit a half-Gaussian profile. Both E→E and I→E connections exhibit clear locality, with characteristic widths of σ ≈ 13–19° for orientation and ≈1 rad for phase. This confirms that neurons with similar preferred features are more strongly connected, consistent with the observed banded structure.

---

> > ### Author Rebuttal · Reviewer_8mru · 2026-04-01
> >
> > Thank you for your response. I read through your other responses as well and think the rebuttal has further improved the paper. I continue to think that this is a good paper that should be accepted to ICML.

---

> > > ### Author Response · Authors · 2026-04-02
> > >
> > > Thank you very much for your time and effort in both reviewing and reading through all the responses! We are also very glad that our rebuttal has addressed your concerns.
> > > Thanks again for the highly insightful and constructive comments, we will include the corresponding changes in the revised manuscript.

---

### Decision · Program_Chairs · 2026-04-30

**Decision:**

Accept (regular)

**Comment:**

This paper studies inhibition-stabilized supralinear networks. While these networks are well-studied in the literature, authors demonstrate that these networks can be trained with biologically plausible learning rules, and the learned network exhibits adaptive coding strategies. This is an important contribution to this field.

Much of reviewer/author discussion focused around clarifying the novelty of the paper's results compared to previous work in the field, and what the actual claim of the paper was. Authors provided a satisfactory framing in their rebuttal, however, this issue needs to be resolved in the final submission. Another important point raised was whether the theoretical analysis provided in the paper actually applies to the trained network. Authors provided a response and a new figure that convinced the reviewer. The AC thinks that this point requires some more exploration, do the response properties of the trained network match what one expects from SSN? Other technical comments were mostly addressed by the authors.

Overall, two out of four reviewers vote for accept, one for weak reject and another for reject. Despite the score averages being borderline, the AC recommends acceptance, but with low priority, given that this is a first-result in training SSN networks.

Additional comment from the AC: Authors claim that "In ICA, neuronal responses are modeled using linear dynamics, with nonlinearity arising exclusively through the learning of synaptic weights", and " Sparse coding extends this framework
by introducing mutual inhibitory interactions between neurons. " These claims are not strictly true. One can have ICA networks with nonlinear neurons and mutual inhibitory interactions. See works on Nonnegative ICA and Hebbian learning. These networks are also trained with biological learning rules.